# Scheduling LLM Inference with Uncertainty-Aware Output Length Predictions

Haoyu Zheng [1]  Yongqiang Zhang [2]  Fangcheng Fu [3]  Xiaokai Zhou [1]  Hao Luo [1]  Hongchao Zhu [2]
Yuanyuan Zhu [1]  Hao Wang [1]  Xiao Yan [4]  Jiawei Jiang [1 5]

## Abstract

To schedule LLM inference, the *shortest job first* (SJF) principle is favorable by prioritizing requests with short output lengths to avoid head-of-line (HOL) blocking. Existing methods usually predict a single output length for each request to facilitate scheduling. We argue that such a *point estimate* does not match the *stochastic* decoding process of LLM inference, where output length is *uncertain* by nature and determined by when the end-of-sequence (EOS) token is sampled. Hence, the output length of each request should be fitted with a distribution rather than a single value. With an in-depth analysis of empirical data and the stochastic decoding process, we observe that output length follows a heavy-tailed distribution and can be fitted with the log-t distribution. On this basis, we propose a simple metric called Tail Inflated Expectation (TIE) to replace the output length in SJF scheduling, which adjusts the expectation of a log-t distribution with its tail probabilities to account for the risk that a request generates long outputs. To evaluate our TIE scheduler, we compare it with three strong baselines, and the results show that TIE reduces the per-token latency by $2.31\times$ for online inference and improves throughput by $1.42\times$ for offline data generation.

## 1. Introduction

Large Language Models (LLMs) underlie many artificial intelligence applications, such as chatbots, content generation, scientific reasoning, and beyond. Popular LLM services

[1]School of Computer Science, Wuhan University, Wuhan, China [2]Dameng Database Co., Ltd., Wuhan, China [3]School of Artificial Intelligence, Shanghai Jiao Tong University, Shanghai, China [4]Institute for Math and AI, Wuhan University, Wuhan, China [5]School of Computer Science, Central China Normal University, Wuhan, China. Correspondence to: Xiao Yan <yanxiao-sunny@whu.edu.cn>, Jiawei Jiang <jiawei.jiang@whu.edu.cn>.

*Proceedings of the $43^{rd}$ International Conference on Machine Learning*, Seoul, South Korea. PMLR 306, 2026. Copyright 2026 by the author(s).

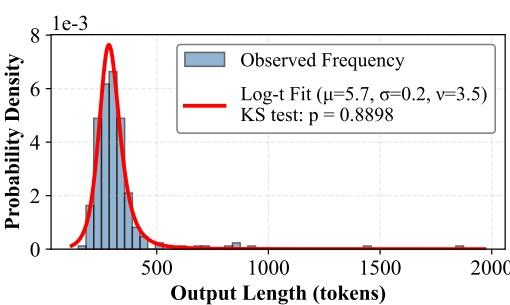

*Figure 1.* Output length distribution for the first prompt in LMSYS-Chat-1M dataset (Zheng et al., 2024). Bars are the output lengths in 256 generations, while red curve is the fitted log-t distribution.

such as ChatGPT (OpenAI, 2022), Gemini (Google DeepMind, 2023), and Claude (Anthropic, 2023) process billions of inference requests on a daily basis (Chatterji et al., 2025). As such, a core problem is how to schedule these inference requests for execution, and the scheduling goals differ according to the target scenarios. In particular, *online serving* (e.g., chatbots and coding assistants) prioritizes latency metrics such as Time-To-First-Token (TTFT) and Per-token Latency (PTLA) for good quality of service (QoS), while *offline processing* (e.g., synthetic data generation (SDG) and data cleaning) emphasizes throughput by maximizing the number of processed requests over a time window.

It has been observed that output length predictions can have large errors (Chen et al., 2025b), and some methods adopt iterative prediction and preemptive scheduling to tackle these errors. For example, TRAIL (Shahout et al., 2025) employs a lightweight linear classifier to predict output length after generating *each* token, preempting requests predicted to generate long outputs. ELIS (Choi et al., 2025) adopts a similar approach but uses an external encoder and extends the prediction interval to 50 tokens. The overheads of these methods can be high due to *frequent predictions and possible preemptions* (vLLM, 2025; Zhong et al., 2024). The limitations of existing methods motivate us to ask:

> *What are the fundamental difficulties in making accurate output length predictions? How to make quality predictions to facilitate scheduling?*

**Our Insights and Solution.** Our key insight is that existing *point estimates* (i.e., predicting an output length for each request) do not match the stochastic decoding process of LLM inference. Specifically, LLM inference randomly samples a token (with probabilities determined by the previous tokens) in each decoding step, and the output length is determined by when the end-of-sequence (EOS) token is sampled. Therefore, the output length of each request is uncertain by nature, and a request can have different output lengths when executed multiple times, as shown in Figure 1. To account for the inherent randomness in the decoding process, the output length of each request should be fitted with a distribution rather than a single value.

By analyzing real requests, we observe that the output lengths follow a *heavy-tailed* distribution that can be effectively fitted using the log-t distribution (Cassidy et al., 2010). We also prove this fact under some mild assumptions on the decoding process of LLM inference. Take Figure 1 for instance, the p-value of the log-t distribution for the Kolmogorov-Smirnov (KS) test (Massey Jr, 1951) is 0.8898, suggesting a high quality fitting. To predict parameters for the log-t distribution of each request, we employ a simple model, which uses fine-tuned DeBERTa-v3-base (He et al., 2021; 2023) to extract request semantics and feed the resultant embedding to an MLP.

To utilize the fitted distribution for scheduling, we devise a metric called *Tail Inflated Expectation (TIE)*. The rationale is that scheduling should account for the risks that requests may generate long outputs and cause HOL blocking, and thus requests with heavy tails should be penalized. Therefore, TIE adjusts the expectation (i.e., $\mathbb{E}\left[X\right]$) of the log-t distribution with its tail expectation (i.e., $\mathbb{E}\left[X \mid X \geq \text{VaR}_\alpha^X\right]$), where $\text{VaR}_\alpha^X$ (Value at Risk) is a tail percentile, to account for the risks. The resulting TIE scheduler is simple to implement as it only replaces output length with TIE for SJF.

We comprehensively evaluate the TIE scheduler for both online and offline inference scenarios and experiment with multiple datasets and models. The results show that compared with the best-performing baseline, TIE reduces the per-token latency by $2.31\times$ for online chatbot and improves throughput by $1.42\times$ for offline SDG tasks. Moreover, micro experiments also suggest that adjusting the expectation with tail in TIE improves performance, and the TIE scheduler generalizes well across different datasets and models.

**Contributions.** We make the following contributions:

- We observe that existing methods make point estimates for request output lengths, and we argue that this does not match the random token sampling of LLM decoding.
- To account for the inherent uncertainty in output length, we propose to fit a log-t distribution instead of a single value. Besides scheduling, our methodology may also benefit other use cases that involve output lengths, such as KV cache management and cost estimation.
- Based on the fitted log-t distributions, we design the TIE scheduler[1], which is simple to implement and provides strong performance in the experiments.

## 2. Related Work

**LLM Serving Systems.** As discussed above, existing LLM systems typically employ FCFS scheduling, which suffers from HOL blocking. Continuous Batching (Yu et al., 2022) is a critical advancement that prevents long-running requests from blocking entire batches. However, queue-level blocking remains an open challenge: long-running requests can still block shorter ones waiting in the queue, leading to increased average latency and reduced system throughput.

**Scheduling for LLM Inference.** To mitigate queue-level blocking, recent studies have explored SJF-based strategies. Beyond SSJF and LTR discussed in Section 1, several other prediction-based scheduling methods have been proposed (Jin et al., 2023; Zheng et al., 2023). However, as discussed earlier, the inherent uncertainty in LLM outputs makes accurate length prediction difficult.

To mitigate this uncertainty, a potential optimization pathway is preemptive scheduling based on iterative prediction (Shahout et al., 2025; Choi et al., 2025; Wu et al., 2024). However, as discussed in Section 1, it incurs substantial overhead. In contrast, our proposed scheduling approach accounts for output uncertainty while avoiding the overhead of frequent re-prediction and preemption.

**LLM Output Length Prediction.** Predicting the output length of LLMs has applications beyond request scheduling, including KV cache management (Horton et al., 2025) and cost estimation (Piotrowski et al., 2025). In addition to prompt-based length predictors such as SSJF and LTR, other methods predict lengths from embeddings and refine predictions iteratively during generation. TRAIL (Shahout et al., 2025) leverages intermediate layer embeddings as input to a linear classifier for predicting output length. Building upon TRAIL, LGR (Piotrowski et al., 2025) models the embeddings from multiple intermediate layers as graph nodes and employs a graph convolutional network to capture inter-layer dependencies for output length prediction. The core limitation of these methods is that point estimates inherently clash with the stochastic nature of autoregressive decoding.

**Stochastic Scheduling.** TIE fundamentally falls within the scope of *stochastic scheduling*, where jobs have random processing times drawn from *known* distributions and the Shortest Expected Processing Time (SEPT) policy (Weber, 1983) minimizes expected flow time. However, several gaps

---

[1]https://github.com/Hyzheng-code/TIE

prevent stochastic-scheduling theory from being directly applied to LLM serving (Mitzenmacher & Shahout, 2025): (i) classical analyses assume the job-size distribution is *fully known*, whereas LLM output lengths must be predicted from the prompt; (ii) most queueing-theoretic results build on the M/G/1 single-server model, while modern LLM serving uses continuous batching; and (iii) practical factors such as prediction overhead, KV-cache contention, and preemption costs further deviate from idealized theoretical settings. Our work serves as a practical step toward bridging this gap by fitting a distribution to each request's output length and incorporating tail-risk-aware scheduling.

## 3. Predicting Uncertain Output Length

The prediction of output length distributions consists of two components: *(1)* selecting and fitting appropriate distributions to model the output length, *(2)* training a predictor to estimate the distribution parameters of output length.

### 3.1. Output Length Modeling via Distribution Fitting

LLM inference is inherently *stochastic*. At each decoding step, the next output token is sampled from a probability distribution conditioned on the preceding context. Therefore, *the same input prompt may yield different outputs*.

**Distribution Selection.** To investigate the distribution of output lengths for identical prompts, we sample 1K prompts from the LMSYS-Chat-1M dataset and generate 100 responses for each prompt. We observe that these distributions exhibit clear *heavy-tailed* characteristics. Specifically, across the 1K distributions, the average *Skewness* $= 3.10$, and the average *Coefficient of Variation (CV)* $= 1.09$, with 78.6% of them $> 1$. The top 10% of output lengths account for 35.7% of the total length, while the average P90/P50 and P99/P50 ratios reach 4.62 and 10.77, respectively.

We now provide a theoretical analysis of this *heavy-tailed* behavior. During LLM inference, tokens are generated autoregressively until the end-of-sequence (EOS) token is produced. Thus, the output length can be expressed as:

$$L = \min\{t \geq 1 : x_t = \texttt{EOS}\}. \quad (1)$$

A key observation is that termination probabilities vary substantially across different generation trajectories, with some trajectories exhibiting low termination rates. For instance, when generating a JSON object, the model maintains a very low termination probability between '{' and '}' to ensure output completeness. To formally characterize this observation, we introduce the following assumption:

**Assumption 3.1.** The termination rate across generation trajectories follows a distribution with density $f$ satisfying: $f(p) \sim c \cdot p^{\alpha-1}$ as $p \to 0^+$ for some constants $\alpha, c > 0$.

*Table 1.* Goodness of fit for common heavy-tailed distributions (1,000 prompts, 100 generations each). KS pass rate: the percentage of prompts where the fit passes the KS test ($p > 0.05$).

| Distribution | # Parameters | KS Pass Rate |
|---|---|---|
| Log-t | 3 | 93.1% |
| Log-t ($\nu = 3.5$) | 2 | 90.6% |
| Log-normal | 2 | 60.3% |
| Gamma | 2 | 32.0% |
| Weibull | 2 | 19.0% |
| Exponential | 1 | 10.7% |

**Theorem 3.2.** *Under Assumption 3.1, the tail probability of output length follows a **power-law decay**:*

$$P(L > n) \sim \frac{c \cdot \Gamma(\alpha)}{n^\alpha} \quad as \; n \to \infty. \quad (2)$$

The detailed proof is provided in Appendix A. The power-law decay established in Theorem 3.2 is widely recognized as a sufficient condition for heavy-tailedness (Clauset et al., 2009; Foss et al., 2011), explaining the high skewness and extreme quantile ratios observed in our empirical analysis.

To identify suitable distribution families, we fit several common heavy-tailed distributions to the empirical distributions obtained above. We assess the goodness of fit using the Kolmogorov-Smirnov (KS) test, where $p > 0.05$ is conventionally considered to indicate adequate fit (Fisher, 1930; D'Agostino & Stephens, 1986; Alasmar et al., 2021).

Table 1 summarizes the parameter count and fitting performance for these distributions. The three-parameter log-t distribution achieves the highest pass rate (93.1%). Since the log-t distribution provides a dedicated parameter $\nu$ to control the tail behavior, we further evaluate two-parameter variants with different fixed $\nu$ values, among which $\nu = 3.5$ yields the best performance (90.6%; detailed results in Appendix D.3). While distribution with more parameters can provide more flexible fits, it also increases the complexity of the prediction model, incurring higher overhead during both training and inference. Balancing efficiency and accuracy, we adopt the variant with fixed $\nu = 3.5$ for subsequent experiments. We present ablation studies comparing end-to-end performance across these distributions in Section 6.4.

**Distribution Fitting.** We focus on the log-t distribution, defined as follows: if $Y \sim t(\nu)$ is a standard Student's t-distribution (Student, 1908) with $\nu$ degrees of freedom, then $X = \exp(\mu + \sigma Y)$ follows a log-t distribution, denoted $X \sim \text{Log-t}(\mu, \sigma, \nu)$. By definition, the probability density function (PDF) of the standard t-distribution is:

$$t_\nu(y) = \frac{\Gamma\left(\frac{\nu+1}{2}\right)}{\sqrt{\nu\pi}\,\Gamma\left(\frac{\nu}{2}\right)} \left(1 + \frac{y^2}{\nu}\right)^{-\frac{\nu+1}{2}} \quad (3)$$

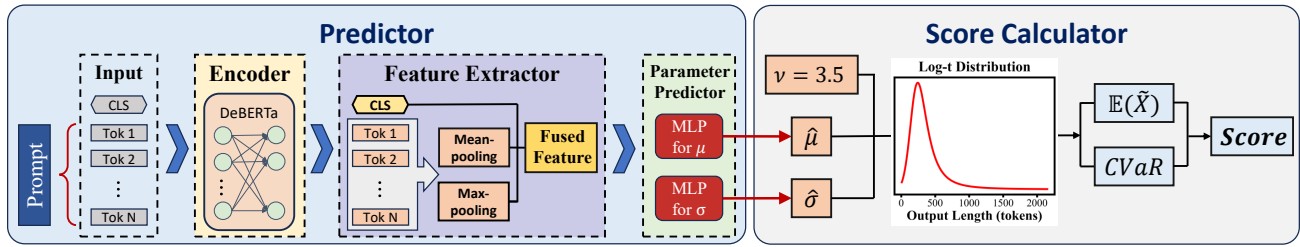

*Figure 2.* Overview of the overall scoring pipeline. The prompt is prepended with a CLS token and encoded by DeBERTa. A multi-pooling strategy aggregates the CLS token, mean-pooled, and max-pooled representations. Two separate prediction heads (MLPs) predict $\hat{\mu}$ and $\hat{\sigma}$, which together with a fixed $\nu = 3.5$ are used to construct the log-t distribution. The final score is computed from $\mathbb{E}(\tilde{X})$ and CVaR.

Accordingly, the PDF of the log-t distribution is given by:

$$f(x \mid \mu, \sigma, \nu) = \frac{1}{\sigma x} \cdot t_\nu \left( \frac{\ln x - \mu}{\sigma} \right) \quad (4)$$

We fit distributions via Maximum Likelihood Estimation (MLE). Given observed data $\{x_1, x_2, \ldots, x_n\}$, the log-likelihood function for the log-t distribution is:

$$\ell(\mu, \sigma, \nu) = \sum_{i=1}^{n} \left[ \ln t_\nu \left( \frac{\ln x_i - \mu}{\sigma} \right) - \ln \sigma - \ln x_i \right] \quad (5)$$

With $\nu = 3.5$, the parameters $\mu$ and $\sigma$ are jointly estimated via the L-BFGS-B optimizer. The optimization objective is:

$$\hat{\mu}, \hat{\sigma} = \arg \max_{\mu \in \mathbb{R}, \sigma > 0} \ell(\mu, \sigma, \nu = 3.5) \quad (6)$$

We fit the observed output length distribution for each prompt using Eq. (6) to obtain the distribution parameters.

### 3.2. Prediction Model for Distribution Parameters

Based on this, we develop a prediction model to estimate the distribution parameters $(\mu, \sigma)$ for each incoming prompt.

**Model Architecture.** As shown in Figure 2, our overall pipeline consists of a predictor (left) and a score calculator (right). The predictor comprises three main components:

**(1) Encoder:** We employ DeBERTa-v3-base (86M backbone) to encode prompts into contextualized embeddings. We select it for its high performance on semantic understanding tasks and efficiency in fine-tuning and inference.

**(2) Feature Extractor:** To capture multi-level semantic information, we employ a *multi-pooling* fusion strategy that combines the `[CLS]` token, *mean-pooled*, and *max-pooled* representations. Specifically, the `[CLS]` token encodes global semantics, mean pooling captures average contextual information, and max pooling highlights salient features.

**(3) Parameter Predictor:** We design separate prediction heads for $\mu$ and $\sigma$. Each head consists of an MLP with three

hidden layers (256, 256, 128) and dropout for regularization. Since the empirical distribution of $\sigma$ is right-skewed, we predict in a transformed space: $\tilde{\sigma} = \log(1 + \sigma)$. **Training Strategy.** We perform z-score normalization on $\mu$ and $\tilde{\sigma}$, and train the model using MSE loss with a *two-stage* strategy: first optimizing all parameters, then *freezing* the encoder and fine-tuning only the prediction heads. This preserves the generalization of the pre-trained encoder. Details are available in our code. The trained model achieves $R^2$ values of 0.82 and 0.76 for $\mu$ and $\sigma$, respectively.

## 4. Scheduling with Tail Inflated Expectation

With the predictor established, we now describe how to leverage the predicted distribution for scheduling.

In stochastic scheduling theory, where job execution times follow known distributions, Shortest Expected Processing Time (SEPT) is a classical strategy (Weber, 1983). However, applying SEPT to our setting faces the following challenges:

1. While the log-t distribution provides a reasonable fit, it cannot perfectly capture the true distribution.

2. The distribution parameters are estimated from the predictor, thereby inevitably containing errors.

3. Requests predicted as long may suffer from starvation.

The first two issues highlight an unavoidable discrepancy between the predicted and actual output lengths. In particular, when a genuinely long request is mispredicted as short, it blocks subsequent requests in the queue. This problem is further exacerbated by the fact that LLM outputs naturally exhibit heavy-tailed behavior. These observations motivate us to *explicitly quantify and manage tail risk in scheduling*.

**Risk-sensitive Scheduling.** We adopt Conditional Value at Risk (CVaR) (Rockafellar et al., 2000) to quantify the tail risk of requests. Intuitively, CVaR represents the expected value conditioned on the tail of a distribution. Formally, for a random variable $X$ and confidence level $\alpha \in (0, 1)$:

$$\text{CVaR}_\alpha(X) = \mathbb{E}\left[ X \mid X \geq \text{VaR}_\alpha(X) \right] \quad (7)$$

where $\text{VaR}_\alpha(X)$ (Value at Risk) denotes the $\alpha$-quantile of $X$. Compared to single-point metrics such as P90, CVaR better characterizes the tail behavior.

Recall that we model output length using the log-t distribution. In practice, LLM generation is forcibly terminated once the output length reaches the `max_tokens` limit. To account for this, we *censor* the predicted distribution at $x_{\max} = \text{max\_tokens}$ (i.e., $\tilde{X} = \min(X, x_{\max})$). For $X \sim \text{Log-t}(\mu, \sigma, \nu)$, define the partial expectation function:

$$\Psi(y) = \int_{-\infty}^{y} \exp(\mu + \sigma s) \cdot t_\nu(s)\, ds, \qquad (8)$$

which represents the cumulative contribution to the expectation of $X$ over the region $\{Y \leq y\}$. The censored expectation and CVaR at confidence level $\alpha$ are then given by (derivations in Appendix B):

$$\mathbb{E}[\tilde{X}] = \Psi(y_{\max}) + x_{\max} \cdot [1 - T_\nu(y_{\max})], \qquad (9)$$

$$\text{CVaR}_\alpha[\tilde{X}] = \frac{\Psi(y_{\max}) - \Psi(y_\alpha) + x_{\max} \cdot [1 - T_\nu(y_{\max})]}{1 - \alpha}, \qquad (10)$$

where $y_{\max} = (\ln x_{\max} - \mu)/\sigma$, $y_\alpha = T_\nu^{-1}(\alpha)$ is the $\alpha$-quantile, $t_\nu(\cdot)$ is the PDF of standard t-distribution (defined in Eq. (3)), and $T_\nu(\cdot)$ denotes its CDF. For scheduling efficiency, we employ *Monte Carlo sampling* with 10k samples to evaluate these metrics instead of numerical integration.

For each request, let $\tilde{X}$ denote the censored predicted output length distribution, we compute a scheduling score as:

$$Score = \mathbb{E}[\tilde{X}] + \beta \cdot \text{CVaR}_\alpha[\tilde{X}] \qquad (11)$$

We set $\alpha = 0.9$ to capture the top 10% tail behavior. Coefficient $\beta$ controls risk sensitivity and is *adaptively* adjusted based on system pressure. Higher pressure means that scheduling a long request would block more subsequent requests for longer periods, thus requiring a more conservative strategy (i.e., a larger $\beta$). Specifically, we measure the system pressure using the waiting queue length $L_q$ and the configured maximum batch size $B$, and adjust $\beta$ within [0.1,0.5] for stability, that is:

$$\beta = \min\left(0.5, \max\left(0.1, \frac{0.1 \cdot L_q}{B}\right)\right) \qquad (12)$$

**Starvation Prevention.** A well-known challenge of SJF-based scheduling is that under high load, long requests tend to be starved by short ones, resulting in excessively long waiting times and compromising scheduling fairness. To mitigate this issue, we introduce a waiting-time decay mechanism that periodically adjusts the scheduling score as $Score' = Score \cdot \gamma^{t_w/\tau}$, where $t_w$ is the waiting time of the request, $\gamma \in (0, 1)$ is the decay factor, and $\tau$ is the

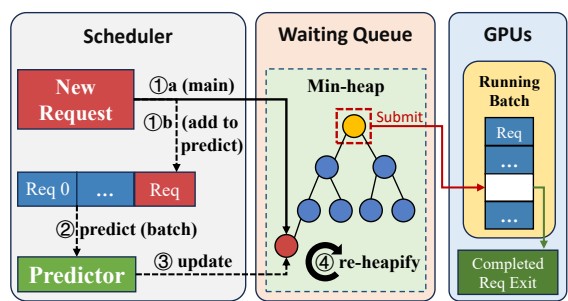

*Figure 3.* Scheduling workflow for new requests. Solid arrows indicate the main workflow, while dashed arrows represent the asynchronous prediction workflow.

decay interval. We set $\gamma = 0.9$ and $\tau = 30s$ by default. This exponential decay ensures that long-waiting requests are gradually prioritized, preventing starvation without disrupting the overall scheduling scheme.

## 5. Implementation and Optimizations

We implement our scheduling strategy on vLLM 0.11.1 (Kwon et al., 2023). In addition, we optimize the deployment to improve the efficiency of prediction and scheduling.

**Challenge.** Existing prediction-based scheduling methods typically employ *synchronous* prediction (Qiu et al., 2024; Shahout et al., 2025), where requests are predicted and sorted by the results before enqueuing. Since requests are blocked during prediction, waiting to form a batch would introduce unacceptable latency. As a result, requests are typically predicted *one at a time*, forgoing the throughput benefits of batching. This is particularly inefficient in the prevalent continuous batching scenario: (1) under low load, continuous batching allows new requests to join the running batch immediately, but synchronous prediction introduces unnecessary blocking; (2) under high load, many requests accumulate while waiting for prediction, and unbatched prediction cannot efficiently identify the shortest request.

To address these issues, we design an *asynchronous* prediction mechanism along with dynamic batching.

**Asynchronous Prediction.** We decouple the scheduler into a *main thread* and a *prediction thread*. The prediction thread is responsible for efficiently predicting pending requests without blocking request enqueuing or the main thread's processing. The main thread, on the other hand, selects a request from the waiting queue to fill a vacant slot in the running batch whenever one becomes available.

Specifically, as illustrated in Figure 3, when the scheduler receives a new request, it inserts the request into the waiting queue (implemented as a *min-heap*) with `max_tokens` as the initial score (①a). This initial score ensures that

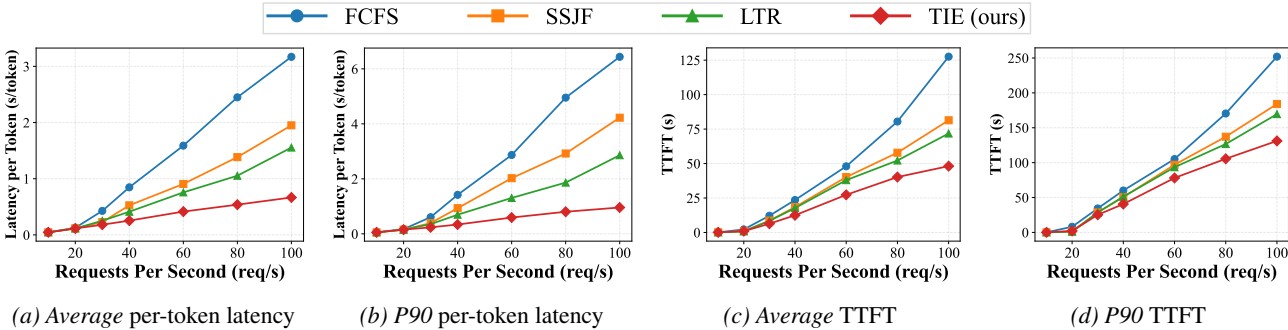

*Figure 4.* Performance of schedulers for online chatbot serving on the real-world workload (LMSYS-Chat-1M) with the 8B model.

unpredicted requests sink to the bottom, allowing already-predicted requests to be executed first, thereby preventing unpredicted long requests from blocking the running batch. Meanwhile, the request is submitted to a prediction queue (①b), where a background prediction thread processes requests *asynchronously* in batches (②). Upon completion of the prediction, the scheduler computes the score and updates the priority of the corresponding request in the heap (③), then re-heapifies to keep the minimum at the top (④).

This design allows new requests to run directly under low load, avoiding unnecessary prediction overhead. In addition, it enables batched prediction for higher throughput.

**Dynamic Batching.** To improve efficiency under varying load conditions, we adopt dynamic batching for the prediction thread. Specifically, requests can wait for a short period to accumulate into a larger batch. Once a timeout (i.e., 3 ms) expires or the request count reaches the maximum batch size (i.e., 32), they are submitted as a batch to the predictor.

Under high load, this batching mechanism allows waiting requests to be predicted rapidly, thereby quickly identifying the shortest request in a long waiting queue.

**Complexity.** Retrieving the minimum element from the waiting queue takes $O(1)$, while insertion, extraction, and priority updates all require $O(\log n)$. This enables the scheduler to efficiently handle requests even under high load.

## 6. Experimental Evaluation

In this section, we compare the proposed TIE scheduling strategy against existing state-of-the-art (SOTA) *non-preemptive* methods. Additionally, we conduct parameter tuning and ablation studies to validate our design choices.

### 6.1. Experiment Settings

**Testbed.** Our experiments are conducted on a server with $8\times$ NVIDIA A6000 (48 GB) GPUs interconnected via NVLink, 128 virtual CPU cores, and 512 GB of memory.

**Serving Models.** In our main experiments, we evaluate two models: Meta-Llama-3-8B-Instruct and Meta-Llama-3-70B-Instruct, both served in FP16 precision. The 70B model is deployed with 8-way tensor parallelism. We also evaluate six models from other families (including three Mixture-of-Experts (MoE) models) in Appendix D.1.

**Workloads and Datasets.** The experimental scenarios encompass both online Chatbot serving and offline Synthetic Data Generation (SDG) tasks. We employ three datasets: LMSYS-Chat-1M (Zheng et al., 2024), ShareGPT (RyokoAI, 2023), and Alpaca (Taori et al., 2023). LMSYS-Chat-1M and ShareGPT both contain *real* user conversations with LLM-based chatbots. Alpaca is a *synthetic* dataset generated via self-instruct using GPT-3.5.

**Baselines.** We evaluate the following methods: (1) **FCFS**, the default strategy in vLLM; (2) **SSJF** (Qiu et al., 2024) and (3) **LTR** (Fu et al., 2024), SOTA *non-preemptive* methods introduced in Section 1; and (4) **TIE** (ours), as described above. Their details are provided in Appendix C.1.

**Training Data.** Unless otherwise specified, the training data for all scheduling methods is derived from the LMSYS-Chat-1M dataset, with outputs generated by Meta-Llama-3-8B-Instruct. Notably, the training data scales are kept *consistent* across methods (900K samples in total): SSJF and LTR utilize the first 900K samples, while TIE uses the first 45K samples with 20 repeated generations per sample (tuned in Appendix D.2) for distribution fitting. This configuration ensures equal training data scale across methods.

### 6.2. Online Chatbot Serving

Following prior work (Fu et al., 2024), we adopt per-token latency (PTLA) and Time-To-First-Token (TTFT) as metrics to reflect user experience in online chatbots. Per-token latency measures the average latency per output token, while TTFT captures the waiting time before the first output token.

**Results.** Figure 4 illustrates the performance of each method under the LMSYS-Chat-1M dataset and the 8B model. As

*Table 2.* Generalization performance of scheduling strategies across different datasets and models under a request rate of 100 RPS. Training data for all methods are derived from LMSYS-Chat-1M using the 8B model.

| Testing Dataset | Testing Model Size | *Average* Per-token Latency (s/token)↓ | | | | *P90* Per-token Latency (s/token)↓ | | | |
|---|---|---|---|---|---|---|---|---|---|
| | | FCFS | SSJF | LTR | TIE | FCFS | SSJF | LTR | TIE |
| LMSYS-Chat-1M | 70B | 9.08 | 5.50 | 4.34 | 2.41 | 16.13 | 8.24 | 7.03 | 4.05 |
| ShareGPT | 8B | 1.66 | 0.95 | 0.86 | 0.57 | 3.27 | 1.84 | 1.56 | 0.90 |
| | 70B | 4.36 | 2.43 | 2.22 | 1.41 | 8.78 | 4.90 | 3.78 | 2.22 |
| Alpaca | 8B | 1.45 | 0.71 | 0.83 | 0.52 | 3.62 | 1.47 | 1.76 | 0.93 |
| | 70B | 4.52 | 2.06 | 2.36 | 1.54 | 11.36 | 4.32 | 5.14 | 2.81 |
| Testing Dataset | Testing Model Size | *Average* TTFT (s)↓ | | | | *P90* TTFT (s)↓ | | | |
| | | FCFS | SSJF | LTR | TIE | FCFS | SSJF | LTR | TIE |
| LMSYS-Chat-1M | 70B | 319.51 | 273.30 | 252.20 | 204.03 | 618.21 | 551.15 | 507.35 | 475.10 |
| ShareGPT | 8B | 161.56 | 132.58 | 120.03 | 98.10 | 316.07 | 296.69 | 273.87 | 239.04 |
| | 70B | 417.28 | 342.45 | 346.50 | 288.79 | 807.27 | 744.59 | 696.75 | 660.28 |
| Alpaca | 8B | 83.65 | 62.93 | 65.33 | 54.18 | 159.72 | 140.80 | 142.17 | 123.02 |
| | 70B | 235.64 | 171.14 | 174.31 | 146.14 | 444.52 | 395.78 | 394.10 | 359.08 |

*Table 3.* Performance of schedulers on SDG tasks, across training datasets and testing model sizes. Evaluated on Alpaca dataset.

| Training Dataset | Testing Model Size | Time for 3k Samples (s)↓ | | | | Throughput within 3 min↑ | | | |
|---|---|---|---|---|---|---|---|---|---|
| | | FCFS | SSJF | LTR | TIE (ours) | FCFS | SSJF | LTR | TIE (ours) |
| LMSYS-Chat-1M | 8B | 229.37 | 144.93 | 139.53 | 98.12 | 2292 | 3488 | 3672 | 4762 |
| | 70B | 559.30 | 324.52 | 316.98 | 246.10 | 861 | 1557 | 1895 | 2524 |
| Alpaca | 8B | 229.37 | 135.79 | 130.67 | 95.08 | 2292 | 3766 | 3663 | 4869 |
| | 70B | 559.30 | 311.06 | 314.36 | 240.19 | 861 | 1634 | 1708 | 2659 |

the requests per second (RPS) increases, all metrics exhibit an upward trend, with FCFS showing the steepest rise. Existing methods (i.e., SSJF and LTR) predict output lengths and sort requests accordingly, reducing per-token latency and TTFT by up to $2.05\times$ and $1.78\times$, respectively. Our proposed TIE further improves upon these results by modeling and predicting output length distributions, thereby leveraging the inherent uncertainty of LLM outputs. As shown in Figure 4a, at 100 RPS, TIE achieves a $4.73\times$ reduction in average per-token latency compared to FCFS, and outperforms SSJF and LTR by $2.91\times$ and $2.31\times$, respectively. Similar results are also observed in TTFT. This improvement stems from our distribution estimation, which comprehensively captures possible output lengths and potential risks.

To evaluate the scheduler's ability to handle workload fluctuations, we examine the increase in average per-token latency when RPS rapidly rises from 30 to 100. For FCFS, SSJF, and LTR, this metric is $7.42\times$, $8.55\times$, and $6.17\times$, respectively; whereas for TIE, it is only $3.68\times$. This improvement is attributed to our risk-adaptive strategy: as requests accumulate in the queue, TIE adopts a more conservative scheduling strategy, deprioritizing potentially long requests, thus reducing the risk of HOL blocking.

**Cross-Model and Cross-Dataset Generalization.** We fur-

ther evaluate the generalization of each scheduling strategy. Note that all predictors are trained on data derived from the LMSYS-Chat-1M dataset using the 8B model. We test them on the ShareGPT and Alpaca datasets, as well as the 70B model. The results are summarized in Table 2.

In terms of per-token latency, on the 70B model, TIE achieves a $3.77\times$ speedup over FCFS, while SSJF and LTR achieve $1.65\times$ and $2.09\times$, respectively. On the ShareGPT dataset, TIE still outperforms existing methods, achieving $2.91\times$ (8B) and $3.09\times$ (70B) speedups over FCFS. Similar results are also observed on Alpaca.

These results demonstrate that TIE generalizes well across diverse settings. This can be attributed to distribution modeling that avoids overfitting to specific workloads, and risk-adaptive scheduling that mitigates prediction errors.

### 6.3. Offline Synthetic Data Generation (SDG)

SDG is emerging as an important inference workload for LLMs due to the scarcity of high-quality data (Chan et al., 2024). In SDG tasks, shorter responses are generally preferred due to their better cost-efficiency, greater diversity, and to mitigate *length bias* in downstream evaluation (Singhal et al., 2024; Dubois et al., 2024). This preference makes

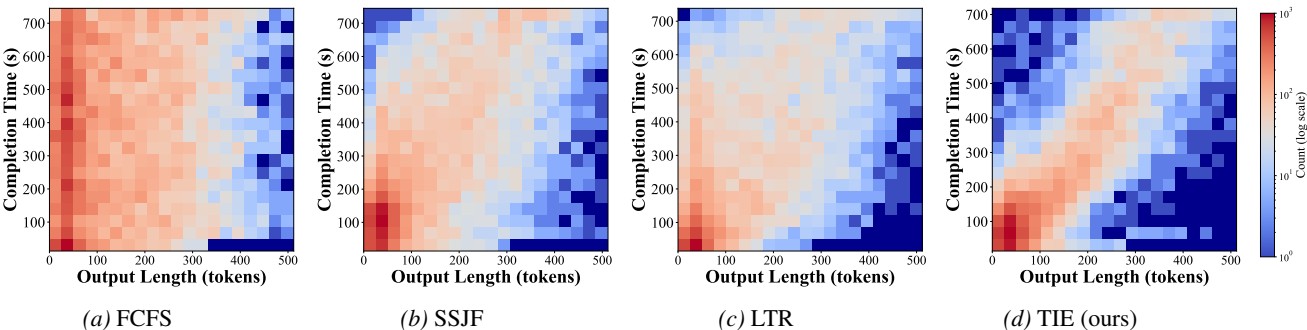

|  | (a) FCFS | (b) SSJF | (c) LTR | (d) TIE (ours) |

*Figure 5.* Heatmaps of completion time versus output length under different strategies on the Alpaca dataset with the 8B model. TIE achieves higher concentration than SSJF and LTR, indicating its ability to accurately capture possible output lengths.

*Table 4.* Ablation study on distribution family and score computation method. Online chatbot experiments use the LMSYS-Chat-1M dataset, and offline SDG experiments use the Alpaca dataset. All experiments are conducted on the Meta-Llama-3-8B-Instruct model. For score computation, CVaR is calculated at $\alpha = 0.9$, and $\beta$ denotes the adaptive risk-sensitivity coefficient. The gray row indicates the default configuration of TIE, orange rows indicate distribution family ablations, and green rows indicate score computation ablations.

| Distribution | Score= | Per-token Latency (s)↓ | | TTFT (s)↓ | | Time for 3k Samples (s)↓ | Throughput within 3 min↑ |
|---|---|---|---|---|---|---|---|
|  |  | Average | P90 | Average | P90 |  |  |
| log-t ($\nu$=3.5) | $\mathbb{E}[X] + \beta \cdot$ CVaR | 0.67 | 0.96 | 48.23 | 131.13 | 98.12 | 4762 |
| log-t (dynamic $\nu$) | $\mathbb{E}[X] + \beta \cdot$ CVaR | 0.69 | 1.02 | 47.76 | 132.40 | 97.70 | 4709 |
| log-normal | $\mathbb{E}[X] + \beta \cdot$ CVaR | 1.63 | 3.37 | 70.91 | 174.88 | 142.21 | 3584 |
| log-t ($\nu$=3.5) | $\mathbb{E}[X]$ *(i.e., SEPT scheduling)* | 0.75 | 1.21 | 52.26 | 145.03 | 108.51 | 4257 |
| log-t ($\nu$=3.5) | $\mathbb{E}[X] + 0.1 \cdot$ CVaR | 0.72 | 1.15 | 50.07 | 141.77 | 104.76 | 4391 |
| log-t ($\nu$=3.5) | $\mathbb{E}[X] + 0.3 \cdot$ CVaR | 0.71 | 1.18 | 49.33 | 139.65 | 105.04 | 4422 |

SDG an important testbed for SJF-based scheduling strategies (Fu et al., 2024). Since the Alpaca dataset is synthetic data generated via self-instruct, it naturally serves as a representative benchmark for SDG workloads (Taori et al., 2023).

Following prior work (Fu et al., 2024), we submit 10K prompts and employ two metrics: (1) the time required to generate 3K samples (i.e., time@3K), and (2) the number of samples generated within 3 minutes (i.e., throughput).

**Results.** As shown in Table 3, SJF and LTR outperform FCFS on both metrics, achieving faster generation speeds, while TIE achieves further improvements. In terms of time@3K, TIE achieves speedups of approximately 2.34×, 1.48×, and 1.42× over FCFS, SSJF, and LTR, respectively. In cross-model and cross-dataset scenarios, although the improvements are somewhat diminished, TIE still maintains strong performance. These results demonstrate the effectiveness of our design: considering the output length of requests from a distributional and uncertainty perspective.

To explain this improvement, we visualize the scheduling patterns of different strategies in Figure 5. The color represents the concentration of requests. Due to space constraints, we only show requests with output lengths less than 512 tokens (over 90% of requests). Full figures for both 8B and 70B models are provided in Appendix D.10.

Under FCFS, requests are distributed almost uniformly across all lengths. SSJF and LTR prioritize shorter requests by predicting output lengths, clustering short requests in the *lower-left* region. However, the clustering becomes weaker for longer outputs, indicating that they struggle to rank requests with longer outputs. This is because longer requests (e.g., "write an article about AI") inherently exhibit higher output uncertainty compared to shorter ones (e.g., "Translate 'thank you' to Spanish"). In contrast, TIE explicitly models the output length distribution and incorporates uncertainty, yielding a more accurate characterization of possible output lengths. As a result, requests scheduled by TIE exhibit a significantly higher concentration, even for those with longer outputs. This result demonstrates that TIE can enhance the effectiveness of the shortest-job-first principle.

### 6.4. Ablation Study

We ablate each component of TIE to assess its contribution.

**Distribution Family.** As discussed in Section 3.1, we evaluate the fitting performance of several common heavy-tailed distribution families and ultimately select the log-t distribution with the fixed $\nu = 3.5$. We also experiment with other distribution families that achieve $> 50\%$ pass rates in the KS test, training models and performing scheduling. Table 4 presents the results of both online and offline experiments.

Compared to TIE, the log-normal distribution, which exhibits inferior fitting performance (60.3% vs. 90.6%), leads to degraded scheduling performance. This indicates that fitting quality directly affects the accuracy of distribution modeling and scheduling effectiveness. The log-t distribution with a dynamic $\nu$ parameter achieves comparable performance to TIE. Considering model complexity and scheduling efficiency, we adopt a fixed $\nu$ parameter.

**Score Computation.** Equation 11 presents our score computation method, which incorporates the expectation, CVaR, and an adaptive risk-sensitivity coefficient $\beta$. We also evaluate alternative score computation methods, as shown in Table 4. Using only the expectation (i.e., SEPT) yields reasonable performance. Incorporating CVaR further improves the results, with the most notable gains achieved when the adaptive risk-sensitivity coefficient $\beta$ is introduced.

We further evaluate $\beta$ across different RPS levels (detailed in Appendix D.7). In brief, the optimal fixed $\beta$ varies with RPS. More importantly, real-world workloads typically exhibit fluctuating RPS, making an adaptive $\beta$ necessary.

**Scheduling Overhead.** We evaluate the overhead of prediction and scheduling. In the online chatbot experiment on LMSYS-Chat-1M with the 8B model at 100 RPS, the average scheduling latency is 4.26 ms per request, while the average TTFT is 48.23 s. Compared to FCFS, which yields an average TTFT of 127.55 s, the overhead introduced by TIE is negligible relative to the performance gains.

## 7. Discussion

**Connection to Learning-Augmented Algorithms.** TIE employs a predictor to estimate distribution parameters, constructs the distribution, and schedules accordingly, thus naturally falling within the *learning-augmented algorithms* framework (Lykouris & Vassilvitskii, 2021; Purohit et al., 2018; Mitzenmacher & Vassilvitskii, 2022). In this framework, the key design challenge is to balance two metrics: *consistency* (good performance when predictions are accurate) and *robustness* (graceful degradation when predictions err). Our scheduling score (Eq. 11) maps directly onto these two metrics: $\mathbb{E}[\tilde{X}]$ captures the prediction-trusting consistency component, reflecting what the scheduler expects the request to take if predictions are accurate, while $\text{CVaR}_\alpha[\tilde{X}]$ captures the robustness component by hedging against tail outcomes arising from prediction errors or inherent stochasticity. The adaptive coefficient $\beta$ (Eq. 12) then governs the trade-off between them: when system pressure is low, $\beta$ remains small and the scheduler favors $\mathbb{E}[\tilde{X}]$ (for consistency); as pressure grows, $\beta$ increases and places more weight on the CVaR term (for robustness). Our ablation (Table 4) confirms this design outperforms both pure SEPT (consistency only) and fixed-$\beta$ variants (static trade-

off), demonstrating that the consistency-robustness principle from learning-augmented algorithms can be effectively realized for LLM inference scheduling.

**Limitations.** A practical limitation of TIE is its training data requirement. Unlike point-estimate predictors, which can be trained directly on production logs from LLM serving systems, TIE requires multiple generations per prompt to fit the output length distribution. We evaluate the impact of training data scales on TIE in Appendix D.5. Although TIE's generalization partially alleviates this concern, addressing the training data requirement (e.g., through few-shot learning) remains an important direction for future work.

## 8. Conclusion

We present TIE, an uncertainty-aware scheduling strategy for LLM inference. We employ the log-t distribution to model the heavy-tailed output length distribution of LLM inference requests, and introduce CVaR with risk-adaptive scheduling to handle the tail risk of output lengths. Experimental results demonstrate that TIE achieves substantial improvements over FCFS and existing SOTA methods in both online and offline scenarios, and exhibits strong generalization across different datasets and models.

## Acknowledgements

This work was sponsored by the National Natural Science Foundation of China (NO. 62472327, No. 62272353), the Key R&D Program of Hubei Province (No. 2023BAB077), the Sichuan Clinical Research Center for Imaging Medicine (YXYX2402), and the Tencent Research Fund. We thank the anonymous reviewers for their constructive feedback, which has helped improve this paper.

## Impact Statement

TIE improves the efficiency and quality of service of LLM inference, and its distribution-modeling perspective may also benefit related areas beyond inference scheduling. As an infrastructure-level optimization, TIE does not introduce new capabilities to the underlying LLMs.

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

# A. Theoretical Analysis of Heavy-Tailed Output Length Distribution

This appendix provides a theoretical analysis supporting Theorem 3.2 in the main text. We show that under Assumption 3.1, the output length distribution exhibits heavy-tailed characteristics with power-law tail decay.

## A.1. Problem Setup

Consider the text generation process of an autoregressive language model. Given an input prompt, the model sequentially samples tokens $x_1, x_2, \ldots$ until generating the end-of-sequence token (EOS). As stated in the main text, the output length is defined as:

$$L = \min\{t \geq 1 : x_t = \text{EOS}\}. \tag{13}$$

Let $p_t = P(x_t = \text{EOS} \mid x_{<t})$ denote the probability of generating EOS at step $t$. Since the preceding context $x_{<t}$ results from stochastic sampling, $p_t$ is itself a random variable that depends on the generation trajectory.

**Lemma A.1** (Tail Probability Expression)**.** *The tail probability of the output length $L$ satisfies:*

$$P(L > n) = \mathbb{E}\left[\prod_{t=1}^{n}(1 - p_t)\right]. \tag{14}$$

*Proof.* The event $\{L > n\}$ occurs if and only if no EOS token is generated in steps $1, 2, \ldots, n$. By the chain rule of conditional probability:

$$P(L > n) = P(x_1 \neq \text{EOS}, \ldots, x_n \neq \text{EOS}) = \mathbb{E}\left[\prod_{t=1}^{n}(1 - p_t)\right], \tag{15}$$

where the expectation is taken over all stochastic generation trajectories. $\square$

For convenience, we restate Assumption 3.1 from the main text.

**Assumption A.2** (Restatement of Assumption 3.1)**.** The termination rate across generation trajectories follows a distribution with density $f$ satisfying $f(p) \sim c \cdot p^{\alpha-1}$ as $p \to 0^+$ for some constants $\alpha, c > 0$.

## A.2. Proof of the Main Theorem

To connect Lemma A.1 with Assumption A.2, we introduce the **effective termination rate** for a generation trajectory:

$$\tilde{p}_n = 1 - \left(\prod_{t=1}^{n}(1 - p_t)\right)^{1/n}, \tag{16}$$

which represents the geometric mean of termination probabilities over the first $n$ steps. This quantity captures the overall tendency of a trajectory to terminate: trajectories generating verbose content (e.g., detailed explanations or structured outputs) tend to have lower $\tilde{p}_n$, while those generating concise responses have higher $\tilde{p}_n$.

In practice, $\tilde{p}_n$ can only be computed retrospectively after generation completes, since each $p_t$ depends on the preceding context due to the autoregressive nature. However, the "type" of a generation trajectory (i.e., whether it will be verbose or concise) is largely determined by early-stage sampling decisions. This motivates Assumption A.2, which posits that each trajectory can be characterized by a fixed termination rate $\tilde{p}$ at the beginning of generation.

Under this assumption, the tail probability in Eq. (14) simplifies to a mixture model:

$$P(L > n) = \mathbb{E}\left[(1 - \tilde{p})^n\right] = \int_0^1 (1 - p)^n f(p)\, dp. \tag{17}$$

We now prove Theorem 3.2 under Assumption A.2.

*Proof of Theorem 3.2.* By Assumption A.2, the density $f$ satisfies $f(p) \sim c \cdot p^{\alpha-1}$ as $p \to 0^+$.

Fix $\delta \in (0,1)$ and decompose the integral in Eq. (17) as:

$$P(L > n) = \underbrace{\int_0^\delta (1-p)^n f(p)\, dp}_{I_1(n)} + \underbrace{\int_\delta^1 (1-p)^n f(p)\, dp}_{I_2(n)}. \tag{18}$$

For $p \in [\delta, 1]$, we have $(1-p)^n \leq (1-\delta)^n$, which yields:

$$I_2(n) \leq (1-\delta)^n \int_\delta^1 f(p)\, dp \leq (1-\delta)^n = O\left((1-\delta)^n\right) = o(n^{-\alpha}). \tag{19}$$

Thus, $I_2(n)$ decays exponentially and is negligible compared to any polynomial decay.

For $I_1(n)$, applying the substitution $u = np$ gives:

$$I_1(n) = \frac{1}{n} \int_0^{n\delta} \left(1 - \frac{u}{n}\right)^n f\left(\frac{u}{n}\right) du. \tag{20}$$

Define $h_n(u) = n^\alpha \cdot (1 - u/n)^n \cdot f(u/n)/n \cdot \mathbf{1}_{[0,n\delta]}(u)$. By Assumption A.2, there exists $M > 0$ such that $f(p) \leq M p^{\alpha-1}$ for all $p \in (0, \delta)$. Since $\delta < 1$, we have $u \leq n\delta < n$ on the domain of integration, so the standard inequality $(1 - u/n)^n \leq e^{-u}$ applies. This gives:

$$|h_n(u)| \leq M \cdot u^{\alpha-1} e^{-u}, \tag{21}$$

where the right-hand side is integrable over $[0, \infty)$ with integral $M\Gamma(\alpha)$. Moreover, for any fixed $u > 0$, as $n \to \infty$:

$$h_n(u) \to c \cdot u^{\alpha-1} e^{-u}. \tag{22}$$

By the dominated convergence theorem:

$$\lim_{n \to \infty} n^\alpha I_1(n) = \int_0^\infty c \cdot u^{\alpha-1} e^{-u}\, du = c \cdot \Gamma(\alpha). \tag{23}$$

Combining the above results, we obtain:

$$P(L > n) = I_1(n) + I_2(n) \sim \frac{c \cdot \Gamma(\alpha)}{n^\alpha} \quad \text{as } n \to \infty. \tag{24}$$

$\square$

## A.3. Implications for Heavy-Tailedness

The power-law tail decay established above has important implications. For any $\lambda > 0$:

$$\lim_{n \to \infty} e^{\lambda n} P(L > n) = \lim_{n \to \infty} \frac{c \cdot \Gamma(\alpha) \cdot e^{\lambda n}}{n^\alpha} = \infty. \tag{25}$$

This is a standard characterization of heavy-tailed distributions (Clauset et al., 2009; Foss et al., 2011), implying that long outputs occur with a non-negligible probability, explaining the high skewness, large coefficients of variation, and frequent occurrence of exceptionally long outputs observed in our empirical analysis (Section 3.1).

*Remark* A.3 (Interpretation of Assumption 3.1 and Assumption A.2). The condition $f(p) \sim c \cdot p^{\alpha-1}$ as $p \to 0^+$ (equivalently, $F(p) \sim \frac{c}{\alpha} p^\alpha$) captures scenarios where a non-negligible fraction of generation trajectories exhibit persistently low termination rates. This occurs naturally in several practical settings. For example, when generating structured outputs such as JSON or code, the model must complete syntactic structures (e.g., matching braces or parentheses) before termination, leading to trajectories with near-zero termination probability over extended spans. Similarly, when the model commits to generating a detailed explanation or a long-form response, it maintains low termination probability until the content is complete. If such trajectories constitute a probability mass proportional to $p^\alpha$ for small termination rates $p$, the heavy-tailed behavior emerges as characterized by Theorem 3.2.

## B. Derivations for the Censored Log-t Distribution

Let $Y \sim t(\nu)$ be a standard $t$-distributed random variable with $\nu$ degrees of freedom. We define the random variable

$$X = \exp(\mu + \sigma Y), \tag{26}$$

which follows a log-$t$ distribution, denoted as $X \sim \text{Log-}t(\mu, \sigma, \nu)$, where $\sigma > 0$ is the scale parameter.

The probability density function (PDF) of the standard $t$ distribution is given by:

$$t_\nu(y) = \frac{\Gamma\left(\frac{\nu+1}{2}\right)}{\sqrt{\nu\pi}\,\Gamma\left(\frac{\nu}{2}\right)} \left(1 + \frac{y^2}{\nu}\right)^{-\frac{\nu+1}{2}}, \tag{27}$$

with cumulative distribution function (CDF) denoted by $T_\nu(y)$. By applying the change of variables $x = \exp(\mu + \sigma y)$, we obtain the PDF of the log-$t$ distribution:

$$f(x \mid \mu, \sigma, \nu) = \frac{1}{\sigma x} \cdot t_\nu\left(\frac{\ln x - \mu}{\sigma}\right), \quad x > 0. \tag{28}$$

We define the censored random variable as $\tilde{X} = \min(X, x_{\max})$, and let

$$y_{\max} = \frac{\ln x_{\max} - \mu}{\sigma}. \tag{29}$$

To simplify subsequent derivations, we introduce the *partial expectation function*:

$$\Psi(y) = \int_{-\infty}^{y} \exp(\mu + \sigma s) \cdot t_\nu(s)\, ds. \tag{30}$$

This function admits a natural probabilistic interpretation: $\Psi(y) = \mathbb{E}[X \cdot \mathbf{1}_{\{Y \leq y\}}]$ represents the contribution to the expectation of $X$ from the event $\{Y \leq y\}$.

In LLM inference, the `max_tokens` parameter imposes an upper bound on the generation length. We therefore censor the distribution at $x_{\max} = \texttt{max\_tokens}$ (i.e., $\tilde{X} = \min(X, x_{\max})$) and compute the censored expectation $\mathbb{E}[\tilde{X}]$, which is always finite. Note that this censoring is necessary because the uncensored expectation $\mathbb{E}[X]$ does not exist for the log-$t$ distribution—the moment generating function of the Student's $t$ distribution is infinite for any $\sigma > 0$ due to its heavy tails.

**Numerical Computation of $\Psi(y)$.** Since $\Psi(y)$ does not admit a closed-form solution, we employ Monte Carlo sampling for efficient computation. Specifically, we draw $N = 10{,}000$ samples $\{Y_i\}_{i=1}^{N}$ from the standard $t$ distribution and approximate:

$$\Psi(y) \approx \frac{1}{N} \sum_{i=1}^{N} \exp(\mu + \sigma Y_i) \cdot \mathbf{1}_{\{Y_i \leq y\}}. \tag{31}$$

### B.1. Censored Expectation

The expectation of $\tilde{X}$ can be decomposed into two terms:

$$\mathbb{E}[\tilde{X}] = \underbrace{\int_{0}^{x_{\max}} x \cdot f(x)\, dx}_{I_1} + \underbrace{x_{\max} \cdot \mathbb{P}(X > x_{\max})}_{I_2}. \tag{32}$$

**Derivation of $I_1$.** Substituting the PDF $f(x) = \frac{1}{\sigma x} \cdot t_\nu\left(\frac{\ln x - \mu}{\sigma}\right)$ and applying the substitution $y = (\ln x - \mu)/\sigma$, we have $x = \exp(\mu + \sigma y)$ and $dx = \sigma \exp(\mu + \sigma y)\, dy$. The integration limits transform from $x \in (0, x_{\max}]$ to $y \in (-\infty, y_{\max}]$,

yielding:

$$
\begin{aligned}
I_1 &= \int_0^{x_{\max}} x \cdot \frac{1}{\sigma x} \cdot t_\nu \left( \frac{\ln x - \mu}{\sigma} \right) dx \\
&= \frac{1}{\sigma} \int_0^{x_{\max}} t_\nu \left( \frac{\ln x - \mu}{\sigma} \right) dx \\
&= \frac{1}{\sigma} \int_{-\infty}^{y_{\max}} t_\nu(y) \cdot \sigma \exp(\mu + \sigma y) \, dy \\
&= \int_{-\infty}^{y_{\max}} \exp(\mu + \sigma y) \cdot t_\nu(y) \, dy = \Psi(y_{\max}).
\end{aligned}
\tag{33}
$$

**Derivation of $I_2$.** Since $X = \exp(\mu + \sigma Y)$, we have $X \leq x_{\max}$ if and only if $Y \leq y_{\max}$. Thus $\mathbb{P}(X \leq x_{\max}) = T_\nu(y_{\max})$, and:

$$
I_2 = x_{\max} \cdot [1 - T_\nu(y_{\max})].
\tag{34}
$$

Combining these results, we obtain:

$$
\mathbb{E}[\tilde{X}] = \Psi(y_{\max}) + x_{\max} \cdot [1 - T_\nu(y_{\max})].
\tag{35}
$$

This result establishes Eq. (9) in Section 4.

## B.2. Censored Conditional Value-at-Risk

The Conditional Value-at-Risk (CVaR) is defined as the conditional expectation beyond the Value-at-Risk (VaR) threshold:

$$
\mathrm{CVaR}_\alpha(X) = \mathbb{E}[X \mid X \geq \mathrm{VaR}_\alpha(X)],
\tag{36}
$$

where $\mathrm{VaR}_\alpha(X) = F^{-1}(\alpha)$ denotes the $\alpha$-quantile.

**Case 1: $\alpha \geq T_\nu(y_{\max})$.** In this case, the $\alpha$-quantile of $\tilde{X}$ equals the censoring point (i.e., $\mathrm{VaR}_\alpha(\tilde{X}) = x_{\max}$), and $\mathbb{P}(\tilde{X} > x_{\max}) = 0$. Since the conditioning event has zero probability, the conditional expectation is not well-defined in the standard sense. Following the convention in risk management literature, we define:

$$
\mathrm{CVaR}_\alpha(\tilde{X}) = x_{\max}.
\tag{37}
$$

**Case 2: $\alpha < T_\nu(y_{\max})$.** The VaR of the censored distribution $\tilde{X}$ is given by:

$$
v_\alpha \triangleq \mathrm{VaR}_\alpha(\tilde{X}) = \exp(\mu + \sigma y_\alpha), \quad \text{where} \quad y_\alpha = T_\nu^{-1}(\alpha).
\tag{38}
$$

By the definition of conditional expectation:

$$
\mathrm{CVaR}_\alpha(\tilde{X}) = \frac{\mathbb{E}[\tilde{X} \cdot \mathbf{1}_{\{\tilde{X} \geq v_\alpha\}}]}{\mathbb{P}(\tilde{X} \geq v_\alpha)}.
\tag{39}
$$

The denominator equals $1 - \alpha$ since $v_\alpha < x_{\max}$. For the numerator, the event $\{\tilde{X} \geq v_\alpha\}$ comprises two cases: (i) $v_\alpha \leq X < x_{\max}$, where $\tilde{X} = X$, and (ii) $X \geq x_{\max}$, where $\tilde{X} = x_{\max}$. Thus:

$$
\mathbb{E}[\tilde{X} \cdot \mathbf{1}_{\{\tilde{X} \geq v_\alpha\}}] = \underbrace{\int_{v_\alpha}^{x_{\max}} x \cdot f_X(x) \, dx}_{J_1} + \underbrace{x_{\max} \cdot \mathbb{P}(X \geq x_{\max})}_{J_2}.
\tag{40}
$$

Following the same substitution as in Section B.1, we obtain:

$$
J_1 = \int_{y_\alpha}^{y_{\max}} \exp(\mu + \sigma y) \cdot t_\nu(y) \, dy = \Psi(y_{\max}) - \Psi(y_\alpha), \quad J_2 = x_{\max} \cdot [1 - T_\nu(y_{\max})].
\tag{41}
$$

Combining these results yields:

$$\text{CVaR}_\alpha(\tilde{X}) = \frac{1}{1-\alpha} \left[ \Psi(y_{\max}) - \Psi(y_\alpha) + x_{\max} \cdot [1 - T_\nu(y_{\max})] \right]. \tag{42}$$

This result establishes Eq. (10) in Section 4.

## C. Experiment Details

### C.1. Baseline Details

Below are the details of the baseline scheduling strategies compared in our evaluation:

- **FCFS (First-Come-First-Served):** The default scheduling policy used in systems like vLLM, which processes requests strictly in arrival order. While ensuring fairness, FCFS suffers from Head-Of-Line (HOL) blocking, where long-running requests delay subsequent shorter ones.

- **SSJF (Speculative Shortest-Job-First) (Qiu et al., 2024):** A scheduling strategy that predicts output lengths using a lightweight proxy model (e.g., fine-tuned BERT) to enable SJF scheduling. It employs a regression head on the [CLS] token representation and prioritizes shorter requests to mitigate HOL blocking.

- **LTR (Learning-to-Rank) (Fu et al., 2024):** A ranking-based scheduling policy that optimizes request ordering based on relative generation lengths rather than exact point estimates. It trains an auxiliary predictor (e.g., OPT-125M) using the ListMLE loss to predict ranking scores that approximate the ideal SJF ordering, with Kendall's Tau used to evaluate ranking quality.

### C.2. Dataset Details

We utilize three distinct datasets to evaluate performance across diverse workloads, including both real-world conversations and synthetic data:

- **LMSYS-Chat-1M (Zheng et al., 2024):** A large-scale real-world LLM conversation dataset collected from the Chatbot Arena. It contains approximately one million conversations involving over 25 different large language models, capturing a wide distribution of user prompts and varying response lengths representative of open-domain chatbot traffic.

- **ShareGPT (RyokoAI, 2023):** A dataset consisting of authentic conversations shared by users from their interactions with ChatGPT. This dataset reflects diverse real-world queries with varying intent and structure, and is widely used to benchmark LLM performance.

- **Alpaca (Taori et al., 2023):** A synthetic dataset generated via the Self-Instruct framework using GPT-3.5. It contains 52,000 instruction-following examples covering a broad range of tasks. Due to its synthetic nature and diversity, it serves as a representative benchmark for Synthetic Data Generation (SDG) workloads.

Figure 6 illustrates the distributions of prompt lengths across these datasets and output lengths on Meta-Llama-3-8B-Instruct and Meta-Llama-3-70B-Instruct (10K samples per dataset).

For prompt lengths, all three datasets exhibit right-skewed distributions with long tails. LMSYS-Chat-1M and ShareGPT show similar patterns, where the mean is approximately three times the median (65.36 vs. 23 and 56.12 vs. 21, respectively), indicating the presence of some exceptionally long prompts. In contrast, Alpaca has considerably shorter prompts with a more concentrated distribution, as reflected in its similar mean and median (13.25 vs 13). This discrepancy arises because Alpaca consists of concise instructional tasks, whereas LMSYS-Chat-1M and ShareGPT contain real-world user queries.

For output lengths, all datasets exhibit pronounced right-skewed distributions with heavy tails, where the mean consistently exceeds the median. Notably, the 70B model tends to produce slightly longer outputs than the 8B model across all datasets, with the most pronounced difference observed on LMSYS-Chat-1M (mean of 203.01 vs 177.19). Among the three datasets, ShareGPT yields the longest outputs, while Alpaca produces the shortest.

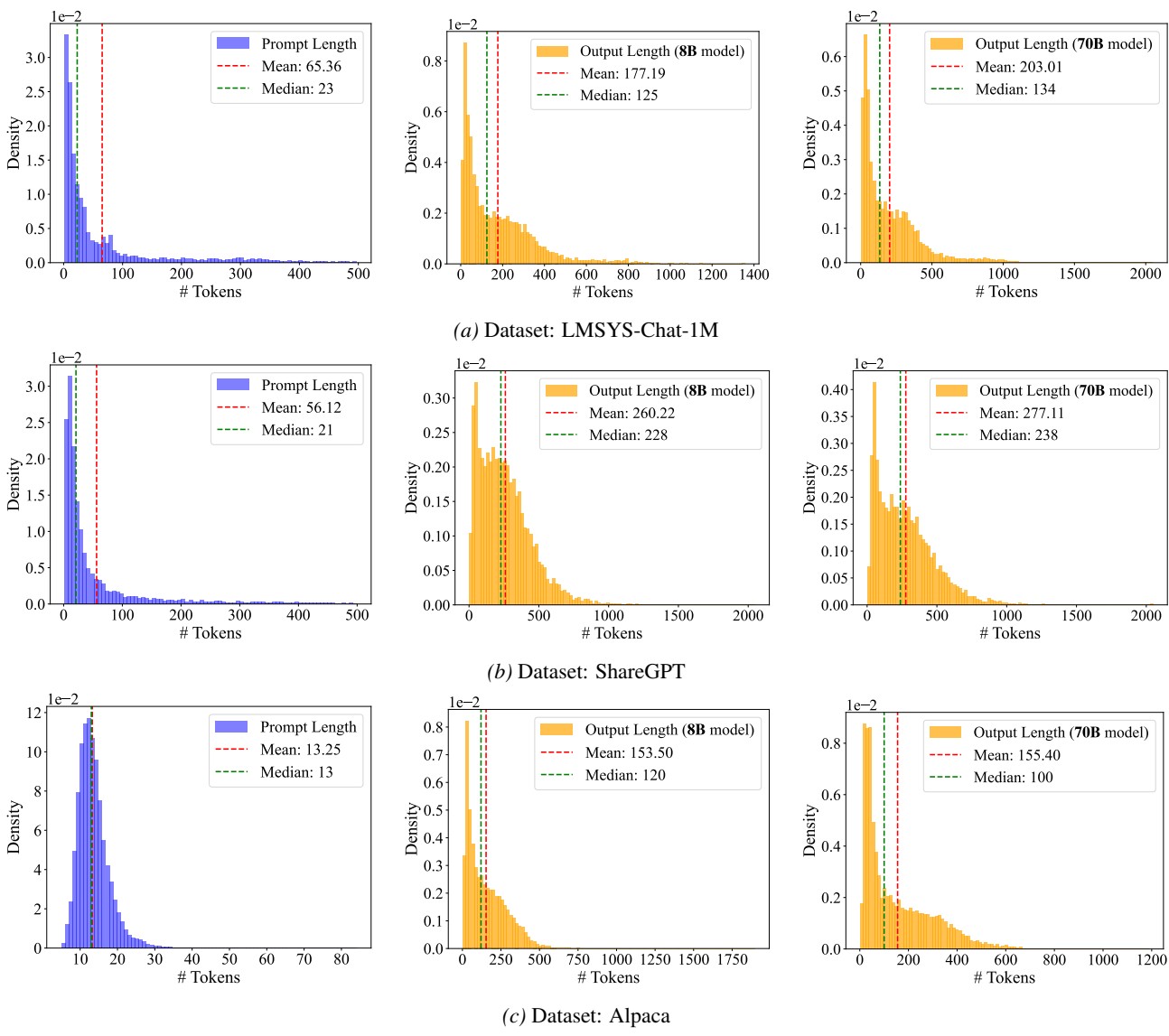

*(a)* Dataset: LMSYS-Chat-1M

*(b)* Dataset: ShareGPT

*(c)* Dataset: Alpaca

*Figure 6.* Distributions of prompt lengths across datasets (tokenized using Meta-Llama-3-8B-Instruct) and output lengths across models.

These observations demonstrate that SDG tasks and Chatbot workloads are fundamentally different in nature, justifying the need to evaluate scheduling strategies under both scenarios separately. Moreover, they highlight a key challenge in LLM inference scheduling: the high variance and heavy-tailed nature of output lengths imply that a small fraction of long-running requests can significantly degrade system throughput and tail latency while blocking subsequent requests (Jiang et al., 2017). This requires schedulers that can accurately identify such requests to ensure Quality of Service (QoS) and system stability (Hu et al., 2019).

## D. Additional Experiments

### D.1. Results on Additional Models

In Section 6, we evaluated scheduling strategies on Meta-Llama-3-8B-Instruct and Meta-Llama-3-70B-Instruct. In this appendix, we evaluate the performance of these strategies on seven additional models from different families to assess their generalizability: GPT-oss-20B (OpenAI, 2025), Qwen3-30B-A3B-Instruct-2507 (Team, 2025), Qwen3-Next-80B-A3B-Instruct (Team, 2025; Yang et al., 2025), DeepSeek-R1-Distill-Qwen-32B (Guo et al., 2025), Mistral-7B-Instruct-v0.3, Mixtral-8x7B-Instruct-v0.1, and OpenPangu-Embedded-7B (Chen et al., 2025a), among which Qwen3-30B-A3B-Instruct-

*Table 5.* Performance comparison of scheduling strategies across different models under the Chatbot workload.

| Model | Dataset | Avg. PTLA (s/token)↓ | | | | Avg. TTFT (s)↓ | | | |
|---|---|---|---|---|---|---|---|---|---|
| | | FCFS | SSJF | LTR | TIE | FCFS | SSJF | LTR | TIE |
| GPT-oss-20B | LMSYS-Chat-1M | 11.55 | 6.77 | 4.96 | 3.00 | 188.33 | 167.68 | 149.46 | 112.34 |
| | ShareGPT | 3.83 | 3.03 | 1.97 | 1.10 | 255.15 | 244.73 | 214.05 | 171.65 |
| | Alpaca | 3.91 | 2.19 | 2.06 | 1.26 | 117.35 | 98.59 | 97.66 | 71.79 |
| Qwen3-30B-A3B-Instruct-2507 | LMSYS-Chat-1M | 8.07 | 3.40 | 2.95 | 1.59 | 150.18 | 109.49 | 96.26 | 73.62 |
| | ShareGPT | 3.76 | 1.75 | 1.50 | 0.86 | 224.70 | 172.47 | 150.54 | 116.26 |
| | Alpaca | 2.58 | 1.14 | 0.82 | 0.49 | 59.72 | 40.82 | 35.55 | 29.88 |
| Qwen3-Next-80B-A3B-Instruct | LMSYS-Chat-1M | 9.60 | 3.53 | 4.51 | 2.86 | 201.17 | 141.52 | 141.35 | 109.97 |
| | ShareGPT | 4.19 | 2.17 | 1.97 | 1.23 | 285.93 | 235.27 | 218.30 | 183.12 |
| | Alpaca | 3.62 | 2.24 | 1.46 | 1.18 | 111.22 | 93.37 | 80.26 | 67.77 |
| DeepSeek-R1-Distill-Qwen-32B | LMSYS-Chat-1M | 2.71 | 1.86 | 1.98 | 1.46 | 1057.02 | 855.83 | 852.08 | 729.42 |
| | ShareGPT | 2.42 | 1.87 | 1.82 | 1.52 | 1443.49 | 1280.86 | 1268.46 | 994.56 |
| | Alpaca | 2.13 | 1.68 | 1.52 | 1.25 | 890.31 | 777.19 | 743.32 | 625.56 |
| Mistral-7B-Instruct-v0.3 | LMSYS-Chat-1M | 2.56 | 1.67 | 1.25 | 0.71 | 127.67 | 100.72 | 88.21 | 59.05 |
| | ShareGPT | 1.19 | 0.82 | 0.79 | 0.57 | 174.43 | 143.29 | 131.63 | 97.71 |
| | Alpaca | 1.34 | 0.94 | 0.81 | 0.47 | 86.13 | 67.72 | 64.18 | 45.04 |
| Mixtral-8x7B-Instruct-v0.1 | LMSYS-Chat-1M | 1.91 | 1.03 | 1.46 | 0.80 | 101.86 | 78.15 | 85.02 | 66.51 |
| | ShareGPT | 1.24 | 0.75 | 0.90 | 0.60 | 137.17 | 114.01 | 117.99 | 94.13 |
| | Alpaca | 1.31 | 0.88 | 0.71 | 0.51 | 76.27 | 63.83 | 59.28 | 42.90 |
| OpenPangu-Embedded-7B | LMSYS-Chat-1M | 2.65 | 0.69 | 1.34 | 0.47 | 163.53 | 106.97 | 113.38 | 93.04 |
| | ShareGPT | 1.64 | 0.57 | 0.75 | 0.36 | 270.92 | 213.25 | 212.58 | 190.33 |
| | Alpaca | 1.46 | 0.47 | 0.55 | 0.30 | 109.42 | 72.27 | 74.92 | 63.47 |

*Table 6.* Performance comparison of scheduling strategies on the SDG task across different models, tested on Alpaca dataset.

| Model | Time for 3k Samples (s)↓ | | | | Throughput within 3 min↑ | | | |
|---|---|---|---|---|---|---|---|---|
| | FCFS | SSJF | LTR | TIE | FCFS | SSJF | LTR | TIE |
| GPT-oss-20B | 283.57 | 257.26 | 202.88 | 165.56 | 1828 | 1957 | 2654 | 3298 |
| Qwen3-30B-A3B-Instruct-2507 (MoE) | 176.27 | 152.69 | 106.53 | 68.80 | 3045 | 3511 | 4563 | 6419 |
| Qwen3-Next-80B-A3B-Instruct (MoE) | 271.22 | 233.51 | 156.85 | 123.91 | 1907 | 2193 | 3379 | 4316 |
| DeepSeek-R1-Distill-Qwen-32B | 2058.48 | 1584.06 | 1524.46 | 1134.33 | 245 | 352 | 394 | 533 |
| Mistral-7B-Instruct-v0.3 | 238.95 | 175.88 | 196.77 | 123.07 | 2246 | 3077 | 2601 | 4173 |
| Mixtral-8x7B-Instruct-v0.1 (MoE) | 259.16 | 201.36 | 144.94 | 105.06 | 2048 | 2619 | 3660 | 5020 |
| OpenPangu-Embedded-7B | 272.33 | 132.13 | 143.07 | 97.60 | 1970 | 3850 | 3608 | 4784 |

2507, Qwen3-Next-80B-A3B-Instruct, and Mixtral-8x7B-Instruct-v0.1 employ MoE architectures. Unless otherwise specified, the experimental setup is identical to that described in Section 6.1.

All models are served with FP16 precision using tensor parallelism. Specifically, Mistral-7B-Instruct-v0.3 and OpenPangu-Embedded-7B are deployed on a single GPU; GPT-oss-20B, Qwen3-30B-A3B-Instruct, and DeepSeek-R1-Distill-Qwen-32B are deployed on 2 GPUs; Mixtral-8x7B is deployed on 4 GPUs; and Qwen3-Next-80B-A3B is deployed on 8 GPUs.

Table 5 presents the performance of different scheduling strategies at 100 RPS in the online chatbot scenario. Table 6 presents the performance in offline SDG tasks. TIE consistently achieves strong performance across all settings, demonstrating its generalizability.

## D.2. Sensitivity to Sample Size

We sample 1k prompts and establish a baseline by fitting distribution parameters using 100 repeated generations per prompt. We then vary the number of repetitions and compute the relative error of the estimated parameters against this baseline.

As shown in Figure 7, the estimation error for $\mu$ remains low even with few repetitions, as the location parameter is primarily determined by the central tendency of the output length distribution, which stabilizes quickly. In contrast, the error in $\sigma$

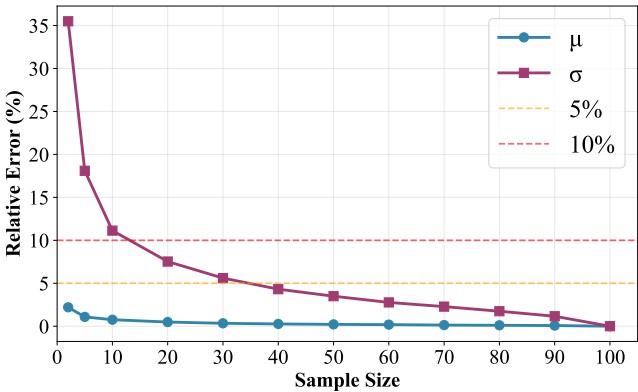

*Figure 7.* Effect of the number of repetitions on parameter estimation accuracy. The relative error is computed against the baseline obtained with 100 repetitions per prompt.

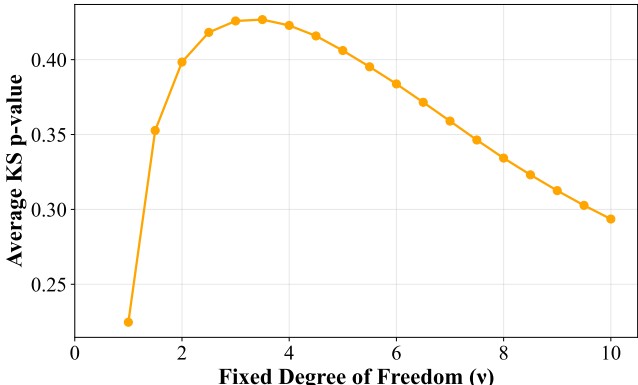

*Figure 8.* Effect of the degrees of freedom parameter $\nu$ on fitting quality.

decreases as the number of repetitions increases, since the scale parameter is sensitive to tail behavior and requires more samples to capture accurately.

Although increasing the number of repetitions yields more precise estimates, it significantly increases the cost of collecting training data for the predictor. We select 20 repetitions as a trade-off between estimation accuracy and data collection cost.

### D.3. Sensitivity to Fixed $\nu$ value

In practice, we use a fixed degree of freedom parameter $\nu$ for fitting. To determine this, we vary $\nu$ from 1 to 10, fit the distribution for each value, and compute the average p-value using the Kolmogorov-Smirnov (KS) test. As shown in Figure 8, $\nu = 3.5$ achieves the best performance.

### D.4. Diagnostic Statistics by Prompt Family

To verify that the heavy-tailed behavior observed in Section 3.1 is not specific to particular prompt types, we classify the $1K \times 100$ prompts into five categories using Qwen3-32B (thinking mode) and analyze their tail statistics. As shown in Table 7, all categories exhibit positive skewness, high P90/P50 ratios, and log-t KS pass rates above 89%, confirming the universality of the heavy-tailed behavior. Open-ended categories exhibit slightly heavier tails than structured ones, consistent with their weaker generation constraints.

*Table 7.* Diagnostic statistics by prompt family for the 1K×100 prompt data (Section 3.1). Prompts classified using Qwen3-32B (thinking mode). Generation model: Llama-3-8B-Instruct. Dataset: LMSYS-Chat-1M. KS Pass: log-t test pass rate at $p > 0.05$.

| Type | Category | #Prompt | Skewness | CV | P90/P50 | P99/P50 | KS Pass |
|---|---|---|---|---|---|---|---|
| Structured | Translation & Reformulation | 266 | 2.75 | 0.93 | 3.40 | 7.80 | **89.5%** |
| | Factual QA & Knowledge | 292 | 3.54 | 1.22 | 4.63 | 11.87 | **93.5%** |
| | Code & Structured Output | 158 | 2.79 | 0.87 | 2.99 | 6.86 | **94.4%** |
| Open-ended | Creative & Open-ended Writing | 165 | 3.42 | 1.12 | 5.60 | 13.00 | **95.7%** |
| | Conversation & Roleplay | 82 | 2.68 | 1.35 | 7.50 | 16.50 | **95.2%** |
| Others | | 37 | 3.00 | 1.45 | 9.50 | 17.50 | **94.0%** |
| **All** | | **1000** | **3.10** | **1.09** | **4.62** | **10.77** | **93.1%** |

*Table 8.* Average per-token latency (s/token) at 100 RPS with different training data scales ($K$ prompts $\times N$ generations each). Model: Llama-3-8B-Instruct. Dataset: LMSYS-Chat-1M.

| Training Data | TIE (ours) | SSJF | LTR |
|---|---|---|---|
| 900K×1 | – | 1.95 | 1.55 |
| 45K×20 | **0.67** | 2.31 | 2.42 |
| 20K×20 | 0.88 | – | – |
| 20K×10 | 1.07 | – | – |
| 15K×10 | 1.58 | – | – |
| 10K×10 | 2.26 | – | – |

## D.5. Sensitivity to Training Data Scale

We vary the predictor's training data scale ($K$ prompts $\times N$ generations each) to evaluate the data efficiency of TIE. As shown in Table 8, with only $15K \times 10 = 150K$ generations (about $1/6$ of the 900K used by SSJF/LTR), TIE already outperforms SSJF and approaches LTR; with the default $45K \times 20$, TIE consistently outperforms both baselines. The multi-generation training cost is incurred only once and can be amortized over the deployment lifetime of an LLM service.

## D.6. Sensitivity to Sampling Temperature

*Table 9.* Average per-token latency (s/token) at 100 RPS. All schedulers trained at temperature=0.7, evaluated at varying temperatures. Model: Llama-3-8B-Instruct. Dataset: LMSYS-Chat-1M.

| Temp. | TIE (ours) | SSJF | LTR |
|---|---|---|---|
| 0.4 | 0.84 | 1.98 | 1.63 |
| 0.7 (as in training) | 0.67 | 1.95 | 1.55 |
| 1.0 | 0.73 | 2.02 | 1.70 |
| 1.3 | 1.13 | 2.59 | 2.62 |
| 1.6 | 2.61 | 4.42 | 5.77 |

To assess generalization across decoding configurations, we train all schedulers at a temperature of 0.7 and evaluate them at temperatures ranging from 0.4 to 1.6. As shown in Table 9, TIE consistently outperforms SSJF and LTR across all temperature settings. At extreme temperatures ($\geq 1.6$), all methods degrade due to severely increased output uncertainty, but such settings are rarely used in practice as they produce low-quality outputs (Nguyen et al., 2025).

## D.7. Effectiveness of Adaptive $\beta$

To validate the adaptive $\beta$ design (Eq. 12), we compare it against fixed $\beta$ values across a range of request rates. As shown in Table 10, the best fixed $\beta$ shifts with load: 0.1 at low RPS, 0.3 at moderate RPS, and 0.5 at high RPS. Since real-world workloads fluctuate over time, no single fixed $\beta$ is optimal. The adaptive scheme achieves the best or near-best performance at every load level by dynamically responding to system pressure.

*Table 10.* Average per-token latency (s/token) with adaptive vs. fixed $\beta$ across request rates. Model: Llama-3-8B-Instruct. Dataset: LMSYS-Chat-1M. **Bold**: adaptive $\beta$; underlined: best fixed $\beta$ per column.

| $\beta$ | 10 RPS | 30 RPS | 60 RPS | 100 RPS | 200 RPS |
|---|---|---|---|---|---|
| **Adaptive** | **0.05** | **0.18** | **0.41** | **0.67** | **1.88** |
| 0.1 | 0.05 | 0.23 | 0.49 | 0.72 | 2.17 |
| 0.3 | 0.07 | 0.20 | 0.44 | 0.71 | 2.09 |
| 0.5 | 0.09 | 0.25 | 0.46 | 0.70 | 1.96 |
| 0.7 | 0.12 | 0.29 | 0.48 | 0.74 | 2.05 |

## D.8. Comparison with Preemptive Scheduling

*Table 11.* Average per-token latency (s/token): TIE vs. TRAIL across RPS. Model: Llama-3-8B-Instruct. Dataset: LMSYS-Chat-1M.

| RPS | 10 | 30 | 40 | 60 | 80 | 100 |
|---|---|---|---|---|---|---|
| TIE (ours) | **0.05** | 0.18 | 0.25 | **0.41** | **0.54** | **0.67** |
| TRAIL | 0.06 | **0.15** | **0.23** | 0.57 | 0.87 | 1.41 |

We adopt TRAIL (Shahout et al., 2025), a *preemptive* scheduler that re-predicts after every generated token, as an additional baseline. As shown in Table 11, the two methods are comparable at low load ($\leq 40$ RPS), with TRAIL holding a slight edge. At moderate and high load ($\geq 60$ RPS), TRAIL degrades sharply as frequent preemptions cause KV-cache evictions and re-prefill overhead. TIE reduces mis-ranking probability at the source through distributional modeling and risk-aware scoring, thereby avoiding the overhead of frequent preemption.

## D.9. Robustness to $\sigma$ Prediction Quality

*Table 12.* Average per-token latency (s/token) under varying $\sigma$ noise levels. $\sigma_{\text{noise}}$: range of multiplicative noise applied to predicted $\sigma$, i.e., the scheduler operates with $(1 \pm \sigma_{\text{noise}}) \times$ predicted $\sigma$. Model: Llama-3-8B-Instruct. Dataset: LMSYS-Chat-1M.

| RPS | TIE ($\sigma_{\text{noise}}$=0) | TIE (0.1) | TIE (0.3) | TIE (0.5) | TIE (0.7) | SSJF | LTR |
|---|---|---|---|---|---|---|---|
| 60 | **0.41** | 0.43 | 0.49 | 0.55 | 0.60 | 0.91 | 0.76 |
| 100 | **0.67** | 0.77 | 1.05 | 1.34 | 1.66 | 1.95 | 1.55 |

We examine TIE's robustness to $\sigma$ prediction errors by multiplying the predicted $\sigma$ with $(1 \pm \sigma_{\text{noise}})$. As shown in Table 12, performance degrades gracefully with noise level. TIE remains competitive with both SSJF and LTR even under $\sigma_{\text{noise}} = 0.7$, indicating that the scheduler is not brittle to imperfect $\sigma$ estimation.

## D.10. Full Heatmaps for SDG Tasks

In this section, we present the full heatmaps for the 8B model (Figure 9) and the 70B model (Figure 10), showing the test results on the Alpaca dataset. By considering the entire output length distribution, TIE leverages richer information and thus achieves more accurate predictions of possible output lengths. This is reflected in the figures as a tighter clustering of requests with similar output lengths.

# E. Future Work

While TIE demonstrates strong performance in scheduling LLM inference requests, several promising directions remain for future exploration:

- **Alternative Distribution Families.** This work adopts the log-t distribution to model output lengths, which effectively captures heavy-tailed characteristics. Future work could explore other distribution families, such as mixture models or non-parametric distributions, to better accommodate diverse output patterns across different applications and domains.

- **Dynamic Distribution Updates During Decoding.** Currently, TIE predicts the output length distribution before decoding begins. An interesting direction is to dynamically update the distribution as tokens are generated, leveraging intermediate decoding states to refine predictions progressively. This could further improve scheduling decisions for requests with high output uncertainty.

- **Online Adaptation for Training Data Collection.** A limitation of this work is that obtaining training data requires generating multiple responses for the same prompt to fit distribution parameters, making it difficult to leverage online serving data directly. Future work could explore online adaptation mechanisms, few-shot learning, or self-supervised approaches to alleviate this constraint and enable continuous model improvement during deployment.

- **Extensions to Other Applications.** Beyond request scheduling, output length distribution prediction has potential applications in other aspects of LLM serving. For example, it could inform KV cache management by pre-allocating memory based on predicted distributions, or enable more accurate cost estimation for API pricing (Jiang et al., 2021). Adapting the distribution prediction framework to these scenarios presents an interesting avenue for future research.

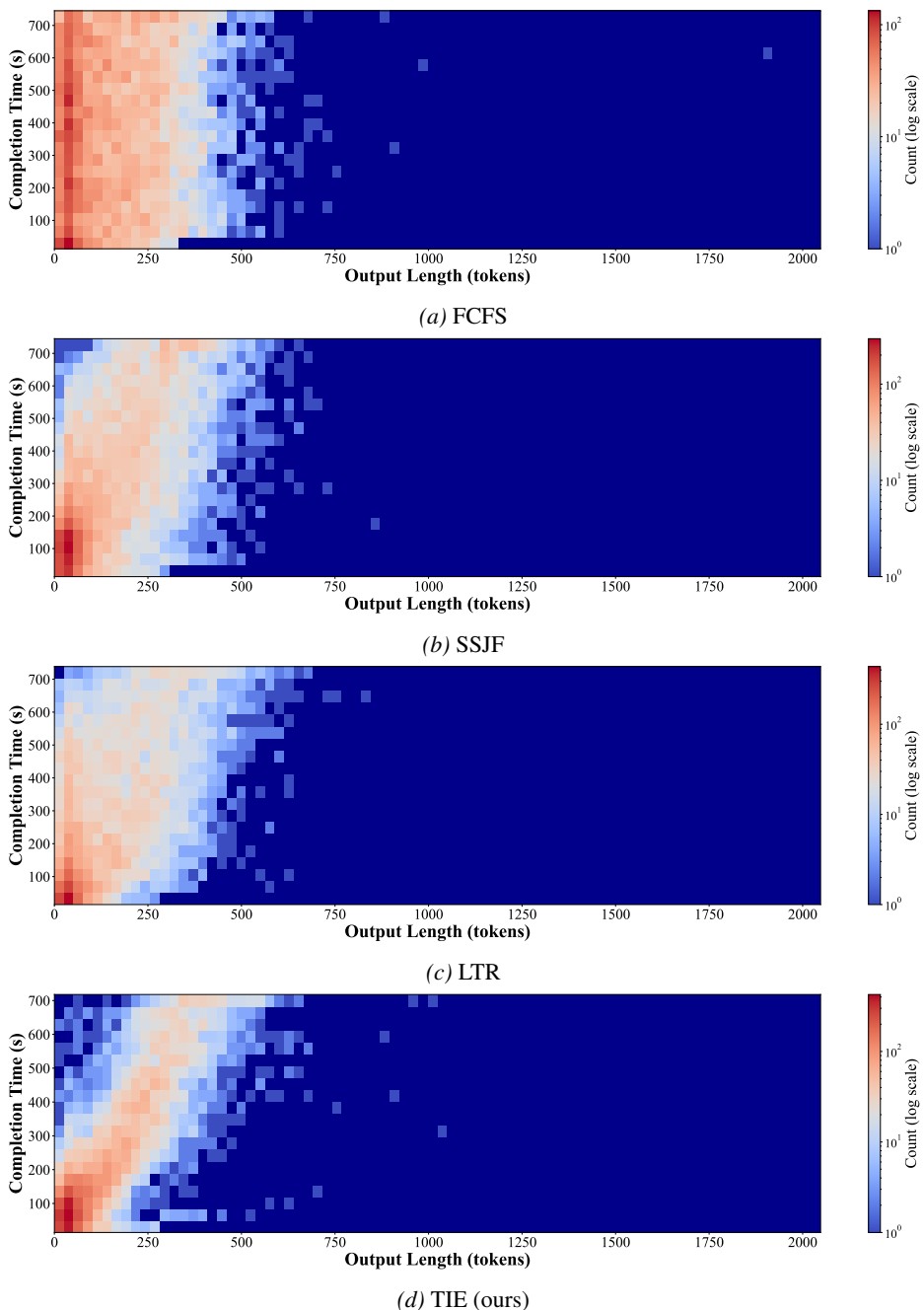

*(a)* FCFS

*(b)* SSJF

*(c)* LTR

*(d)* TIE (ours)

*Figure 9.* Full heatmaps of completion time versus output length under different strategies on the Alpaca dataset with the 8B model.

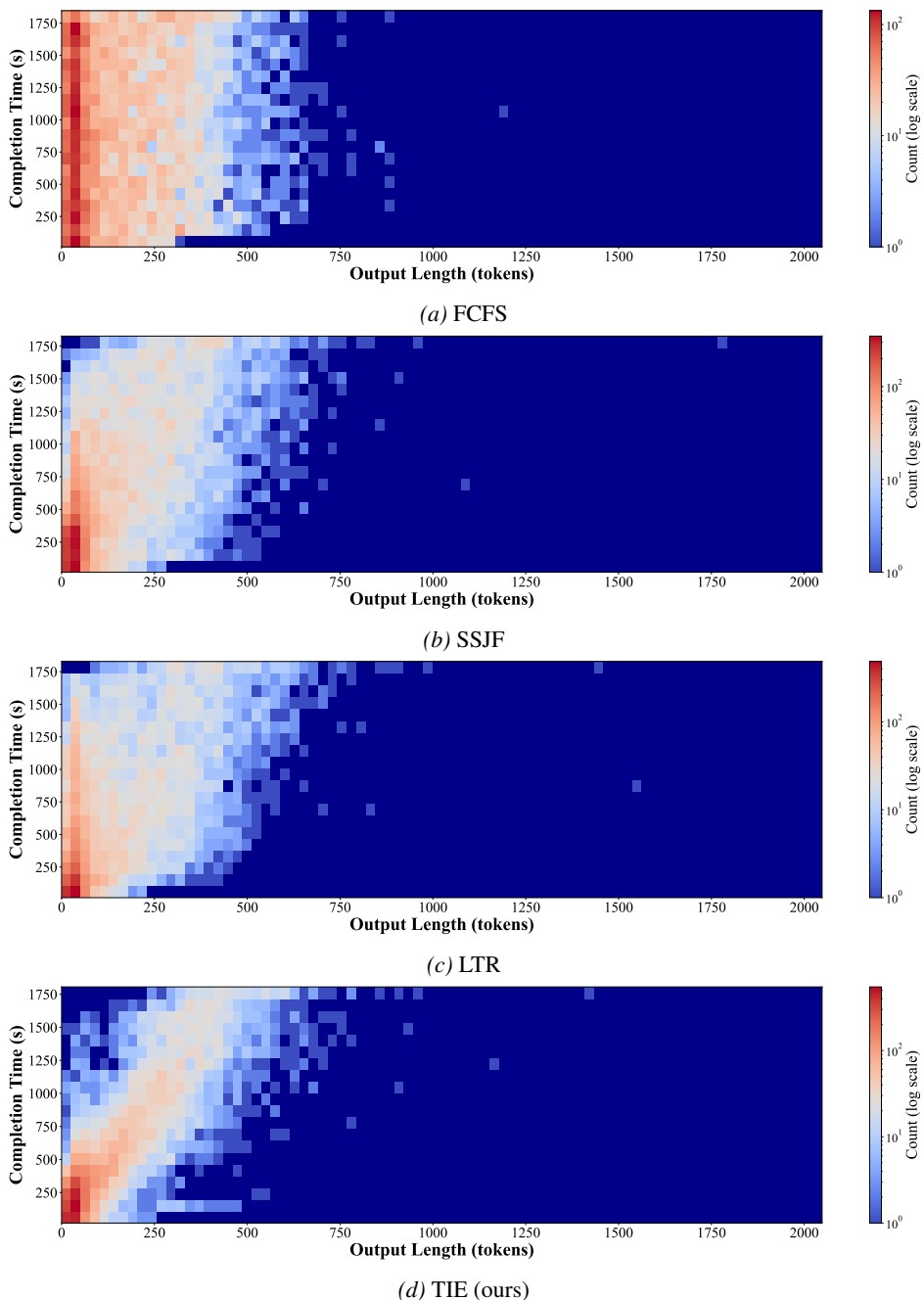

*(a)* FCFS

*(b)* SSJF

*(c)* LTR

*(d)* TIE (ours)

*Figure 10.* Full heatmaps of completion time versus output length under different strategies on the Alpaca dataset with the 70B model.

