# OpenReview forum: "Scheduling LLM Inference with Uncertainty-Aware Output Length Predictions"
_ICML.cc/2026/Conference — ICML 2026 regular_

### Official Review · Reviewer_CKAc · 2026-02-13

**Soundness:** 3
**Presentation:** 3
**Significance:** 4
**Originality:** 3
**Overall Recommendation:** 5
**Confidence:** 3

**Summary:**

The paper studies the problem of LLM inference scheduling:
prompts arrive and must be scheduled on an inference platform
without preemption, that is, as soon as a prompt (job) started,
it must be run to completion. The goal is to minimize the
average latency over all jobs. If the duration of an
inference run of a job is known, which for simplicity can be thought of
being proportional to the number of output tokens, the optimal
strategy if all jobs are available from the beginning is to
schedule jobs by increasing length. However, a
scheduler does not know a jobs length before the model returns
the end-of-sequence token. Existing schedulers therefore do
simple algorithms like First-Come-First-Serve, which have bad
worst-case performances, or try to predict the length of the
inference run based on the prompt, which can be an arbitrarily bad estimate
given the highly stochastic nature of LLM inference.

The present paper proposes to predict a
distribution for the length of an inference run based on
the prompt and the model. Specifically, the authors argue that under certain
assumptions on the stochastic process of token generation,
the length follows a log-t distribution. Such distributions can be described with few predicted parameters.

Based on this model, the authors implement a predictor, which outputs a length distribution for each prompt, and
a scheduler, which takes the predicted distribution and
schedules the inference runs based on an extended variant
of the classic Shortest Expected Processing Time strategy.

The empirical results show a major improvement in latency over FCFS and over other SOTA baselines that predict only a single length
value of the inference run.
They also demonstrate strong generalization across different datasets and models of their predictor.
Moreover, their analysis shows that
their algorithm's latency are proportional to their actual output length, which means that the scheduler is quite effective.

**Compliance With Llm Reviewing Policy:**

Affirmed.

**Final Justification:**

I think this is a strong paper that should be accepted. The promised revision by the author will increase its positioning and visibility.

**Key Questions For Authors:**

I am mainly interested in whether how this relates to stochastic scheduling / queuing and algorithms with predictions / learning-augmented algorithms. Showing a stronger connection to these area would further improve the paper.

**Limitations:**

yes

**Strengths And Weaknesses:**

### Strengths
- Relevance: There is not much to argue that LLM inference
scheduling is currently a super important problem, and will continue to be so in the (near) future.
- New models: the authors propose an improved approach of how to model predictions for inference runs.
- Full approach: the paper gives both a predictor and a scheduler. Hence, the paper feels complete and a full package.
- Strong empirical results: the experiments are extensive
and the results show a significant improvement over SOTA.
- The paper is generally well written and the story is easy to follow.


### Weaknesses
- The paper gives only empirical results, but no theoretical performance guarantees.
- I am missing more details on the connection to scheduling / queuing theory, in particular on stochastic scheduling, where it is usually assumed that distributions over job sizes are known. Also, the paper fits will in the ''algorithms with predictions'' framework (Mitzenmacher & Vassilvitskii '22), where an algorithm (the scheduler) is given a prediction on uncertain data. The paper is a prime example of this, but there are no references to this subarea. Perhaps this can be explored in the rebuttal.


### Minor comments:
- Line 290 right: whitespace missing

---

> ### Author Rebuttal · Authors · 2026-03-31
>
> Thank you for your recognition of our work and your constructive concerns. We address your questions below.
>
> **Q1+W2 (a). Connection to stochastic scheduling and queueing theory.**
>
> **Scope.** TIE models output lengths with log-t distributions and performs shortest-job-first-like scheduling accordingly, placing it within the scope of both stochastic scheduling and queueing systems. However, although these fields have been extensively explored, applying existing results to LLM inference scheduling remains challenging. As Mitzenmacher & Shahout(2025) \[1\] point out, this requires addressing a series of open problems, such as resource contention, KV cache reuse and eviction, and prediction overhead (when a predictor is involved).
>
> **Gap & Positioning.** Specifically, classical stochastic scheduling typically assumes that job size distributions are fully known, whereas in LLM inference, the actual output length during decoding does not strictly follow any known distribution. Our work serves as a practical effort to **bridge this gap** by combining distributional modeling of output lengths with a risk-adaptive scheduling strategy tailored to the unique characteristics of LLM serving.
>
> **Q1+W2 (b). Connection to the algorithms with predictions framework / Learning-augmented algorithms.**
>
> **How TIE fits.** We agree that TIE naturally falls within this paradigm: the predictor provides estimated distribution parameters, and the scheduler must maintain a trade-off between the cases where predictions are accurate *(consistency)* and where predictions contain errors *(robustness)*. Specifically, in our design, the introduction of CVaR and the adaptive risk-sensitivity coefficient β (Eq. 12\) **explicitly govern this trade-off**: when system pressure is low, β remains small, and the scheduler relies more heavily on the predicted expectation *(favoring consistency)*; when pressure increases, β grows larger, placing greater weight on the CVaR term *(favoring robustness)*. The ablation study empirically validates that this design outperforms both pure SEPT (consistency only, no robustness) and fixed-β  variants (static trade-off regardless of system conditions).
>
> **Existing and broader connections.** In fact, the baselines discussed in our paper (e.g., SSJF and LTR) themselves fall within the learning-augmented algorithms framework. We will make this connection explicit in the revised paper and further discuss more general learning-augmented scheduling methods to better situate our work within this broader literature.
>
> In the Related Work section of our manuscript, due to space constraints, we focused on work directly related to LLM inference scheduling to facilitate understanding and comparison, and briefly mentioned stochastic scheduling in Section 4\. In the revision, **we will discuss these related fields and clarify our positioning.**
>
> **W1. Theoretical performance guarantees.**
>
> **Practical complexity.** As discussed above, substantial gaps exist between practical LLM inference serving and the classical theory of stochastic scheduling and queueing. For instance, the widely adopted **M/G/1 model** assumes that requests are served one at a time, whereas continuous batching in practice processes hundreds of requests simultaneously during decoding. These discrepancies make it **difficult to precisely characterize** the theoretical performance of the proposed scheduler, as this would likely require highly complex parametric analysis along with knowledge of the true output length distributions. For the same reason, prior work on LLM inference scheduling (e.g., SSJF, LTR) likewise provides no theoretical performance guarantees.
>
> **Our theoretical contribution.** That said, we do provide theoretical analysis in Appendix A, proving that under mild assumptions, LLM output lengths follow a heavy-tailed distribution. We believe distributional estimation is a promising research direction, as it better aligns with the stochastic decoding process of LLMs, and our heavy-tailed analysis can serve as a valuable reference for future work.
>
> We will incorporate the above analyses in the revision. Thank you for your recognition and valuable suggestions\!
>
> \[1\] Mitzenmacher, M., & Shahout, R. (2025). Queueing, predictions, and large language models: Challenges and open problems. *Stochastic Systems*, *15*(3), 195-219.

---

> > ### Author Rebuttal · Reviewer_CKAc · 2026-03-31
> >
> > I thank the authors for their rebuttal.
> >
> > I will keep my assessment that this is a good paper that should have a chance of being accepted.

---

> > > ### Author Response · Authors · 2026-04-07
> > >
> > > We sincerely thank the reviewer for the positive assessment and for the valuable suggestions.
> > >
> > > We will incorporate a more detailed discussion on the connections to stochastic scheduling / queuing and learning-augmented algorithms to better clarify the positioning of our work in the revised paper.

---

### Official Review · Reviewer_J814 · 2026-02-24

**Soundness:** 4
**Presentation:** 3
**Significance:** 4
**Originality:** 3
**Overall Recommendation:** 5
**Confidence:** 2

**Summary:**

This paper analyzes the limitations of three scheduling approaches: First-Come-First-Served (FCFS), Shortest Job First (SJF), and iterative prediction paired with preemptive scheduling. To address these issues, the authors propose the Tail Inflated Expectation (TIE), an uncertainty-aware scheduling framework. First, it precisely models the heavy-tailed distribution characteristics of LLM output lengths using the log-t distribution, overcoming the limitations of traditional point estimation paradigms. Second, it introduces Conditional Value at Risk (CVaR) to quantify the tail risk of long outputs, and designs the TIE scheduling score metric that integrates distribution expectation and tail risk to achieve risk-adaptive scheduling decisions. Meanwhile, the framework is complemented with an asynchronous prediction mechanism and dynamic batching strategy, which effectively reduces scheduling overhead and significantly improves system throughput under high load.
Extensive experimental results demonstrate that the proposed method significantly outperforms existing state-of-the-art approaches in both online and offline scenarios, and exhibits robust generalization across various models and datasets.

**Compliance With Llm Reviewing Policy:**

Affirmed.

**Final Justification:**

My recommendation is Accept.The authors have addressed my major concerns, and the core contributions of this paper are clear and distinct, with considerable practical value.

**Key Questions For Authors:**

1. In this paper, a parameter predictor is employed to estimate μ and σ, and the model learns by maximizing the likelihood of the training data during training. If there exists a distribution shift problem between real-world data and training data (i.e., when the distribution of real-world data differs significantly from that of the training data), how should this scenario be addressed?

**Limitations:**

yes

**Strengths And Weaknesses:**

Soundness: This paper exhibits outstanding technical reliability, with its arguments supported by both solid theoretical foundations and sufficient empirical validation. Theoretically, it completes the mathematical proof of the heavy-tailed characteristics of output lengths based on reasonable assumptions, and the log-t distribution fitting is validated by the Kolmogorov-Smirnov (KS) test (pass rate of 93.1%), forming a complete logical chain. The experimental design is rigorous, covering dual scenarios (online and offline), multiple datasets, and various models, while controlling for consistency in training data scale to ensure reproducible results. The authors quantify the contributions of each component through ablation studies, honestly report scheduling overhead and data collection limitations, reflecting an objective and sincere evaluation attitude.
Presentation: This paper has a clear structure, with core mechanisms progressing in a hierarchical manner. Its expression is professional and standardized, with consistent use of terminology; formulas and figures intuitively support key arguments, balancing academic depth and readability. The literature review of related work is comprehensive, accurately distinguishing the differences between this work and existing methods, and providing sufficient information for reproduction with precise positioning in the literature.
Significance: This paper focuses on the core pain points of LLM inference scheduling, addressing critical issues such as First-Come-First-Served (FCFS) blocking and the mismatch between point estimation and stochastic decoding, demonstrating prominent practical value. It achieves significant performance improvements: reducing online inference latency by 2.31× and increasing offline throughput by 1.42×. Moreover, it is implemented based on vLLM, enabling easy deployment. The distribution modeling approach can be extended to fields such as KV cache management and cost estimation, embodying both research extension value and practical application potential.
Originality: This paper breaks the traditional paradigm of point estimation for output length, proposing a novel scheduling framework of "distribution modeling + risk quantification" and being the first to adapt the combination of log-t distribution and Conditional Value at Risk (CVaR) to LLM scheduling. It integrates three core modules, including distribution prediction, risk-adaptive scoring, and asynchronous batching. Through large-scale experiments, it verifies the effectiveness of distribution modeling, corrects traditional cognitive biases, and opens up a new direction of uncertainty-aware modeling for the field of LLM inference scheduling.

---

> ### Author Rebuttal · Authors · 2026-03-31
>
> We are delighted that our work has received your recognition, and we sincerely appreciate your positive evaluation!
>
> In the manuscript (Table 2, Table 3, and Appendix D.1), we have discussed the performance of the prediction model trained solely on the LMSYS-Chat-1M dataset with Llama-3-8B-Instruct, evaluated across 3 datasets and 8 models (including MoE architectures). TIE exhibits strong generalization capability. This is because the semantics of the prompt remain a dominant factor in determining the output length, even under different datasets and models. Meanwhile, we mitigate HOL blocking through distributional estimation and a tail-risk-aware score computation method, and achieve more conservative scheduling under high RPS by adaptively adjusting the sensitivity coefficient β.
>
> To further demonstrate the robustness of TIE, we simulate two scenarios where the training data distribution deviates from the actual distribution: (1) Since σ is inherently more difficult to predict than μ (R^2 \= 0.76 vs. 0.82), we inject noise into the predicted σ to simulate inaccurate estimation of this parameter; (2) We vary the decoding temperature during inference to simulate a distributional mismatch between training and inference conditions.
>
> All experiments below are conducted on Llama-3-8B-Instruct with the LMSYS-Chat-1M dataset, reporting average per-token latency (s/token) at 100 requests per second.
>
> **Case1: Robustness to σ Prediction Quality**
>
> We conduct a controlled noise experiment, which shows that the performance degradation caused by σ prediction errors is **gradual**.
>
> **Table 1\.** Avg Pt-Lat under varying σ noise levels.
>
> | RPS | TIE ($σ\_{noise}$=0) | TIE (0.1) | TIE (0.3) | TIE (0.5) | TIE (0.7) | SSJF | LTR |
> | :---- | :---- | :---- | :---- | :---- | :---- | :---- | :---- |
> | 60 | **0.41** | 0.43 | 0.49 | 0.55 | 0.60 | 0.91 | 0.76 |
> | 100 | **0.67** | 0.77 | 1.05 | 1.34 | 1.66 | 1.95 | 1.55 |
>
> At moderate load (60 RPS), adaptive β buffers the tail's influence, keeping performance stable. At high load (100 RPS), only under extreme noise (0.7×, i.e., scheduling with 0.3–1.7× of the predicted σ) does TIE (1.66) fall within the range of baselines. Note that this threshold (0.7×) is relative to predicted values; accounting for inherent prediction error, the tolerance relative to ground-truth is even larger.
>
> The scale parameter σ captures output length **variability** for a given prompt, which is inherently harder to predict than the location parameter μ (reflecting the approximate output length). Nevertheless, the above results demonstrate that controllable prediction errors in σ have a limited impact on end-to-end performance.
>
> **Case2. Sensitivity to Sampling Temperature**
>
> Temperature controls sampling randomness and thus affects output length dispersion. However, the relative ordering among requests is largely preserved: prompts that tend to produce longer outputs still do so. We show that TIE trained under a temperature of 0.7 **generalizes well** across practical temperature settings:
>
> **Table 2.** Avg per-token latency (s/token) across sampling temperatures. All models are trained at temperature=0.7 and evaluated at test-time temperatures from 0.4 to 1.6.
>
> | Temp. | TIE (ours) | SSJF | LTR |
> | :---- | :---- | :---- | :---- |
> | 0.4 | 0.84 | 1.98 | 1.63 |
> | 0.7 (as in training) | 0.67 | 1.95 | 1.55 |
> | 1.0 | 0.73 | 2.02 | 1.70 |
> | 1.3 | 1.13 | 2.59 | 2.62 |
> | 1.6 | 2.61 | 4.42 | 5.77 |
>
> (1) At low temperature (≤0.4), reduced uncertainty dilutes distributional modeling's advantage over point estimation, but TIE still leads. (2) At moderate temperature (0.7–1.3), TIE performs best as dispersion remains within a manageable range. (3) At extreme temperature (≥1.6), all methods degrade due to severely increased uncertainty and frequent HOL blocking. However, such settings are rarely used in practice as they produce low-quality outputs \[1\].
>
> In summary, TIE **generalizes well** across practical temperature settings, while baselines (SSJF, LTR) are more brittle under high-temperature settings due to the lack of risk awareness.
>
> We hope the above experiments adequately address your concerns. We sincerely thank you again for your recognition\!
>
> \[1\] Minh et al. Turning Up the Heat: Min-p Sampling for Creative and Coherent LLM Outputs. ICLR 2025\.

---

> > ### Author Rebuttal · Reviewer_J814 · 2026-04-01
> >
> > Thanks to the authors for their explanations.

---

> > > ### Author Response · Authors · 2026-04-07
> > >
> > > We thank the reviewer for the positive feedback and for confirming that our explanations have addressed the concerns.

---

### Official Review · Reviewer_9d7q · 2026-03-08

**Soundness:** 3
**Presentation:** 3
**Significance:** 2
**Originality:** 2
**Overall Recommendation:** 4
**Confidence:** 4

**Summary:**

This paper addresses scheduling of LLM inference requests by proposing to model output length as a distribution rather than a point estimate. The classical Shortest-Job-First (SJF) algorithm predicts is based on single point estimates, which doesn't match the stochastic nature of LLM decoding. The authors focus on a general theme of uncertainty-aware scheduling by fitting output lengths to a log-t distribution and introducing the Tail Inflated Expectation (TIE) metric, which combines expected value with Conditional Value-at-Risk to account for tail risks. The method is evaluated on online chatbot serving and offline synthetic data generation, showing improvements over FCFS, SSJF, and LTR baselines (2.31× reduction in per-token latency, 1.42× throughput improvement). The paper includes theoretical analysis of heavy-tailed output distributions and implements asynchronous prediction with dynamic batching.

**Compliance With Llm Reviewing Policy:**

Affirmed.

**Final Justification:**

As mentioned in my initial review, the main concern why I recommended rejection of the paper was the lack of comparison with stochastic and learning-augmented scheduling algorithms, which are very relevant to the context of the current paper. The authors explained in their rebuttal the connection to these fields and why comparing to such algorithms is difficult.
Furthermore, I had other concerns about the design choices (why CVaR?, why adaptive $\beta$?) that were addressed by the authors as they provided additional experimental results comparing their choices to other natural ones.
For these reasons, I raise my score to "Weak accept". Not to "Accept" because the overall contribution of the paper, although very interesting, is not strong enough for clear acceptance.

**Key Questions For Authors:**

1. Can the authors address the weakness 2 on the specific choice of TIE?
2. The specific design choice for the parameter $\beta$ is a bit unclear:
2.a) Can the authors explain more the choice of the adaptive formula for parameter $\beta$?
2.b) Why was it adjusted specifically in [0.1,0.5]? for e.g. why not something wider like [0.05,0.5] or tighter like [0.2,0.4]. The authors explain it is adjusted for "stability", can they explain more what this means? How sensitive is this stability to the adjustment of $\beta$?
2.c) Table 4 reports results for constant $\beta = 0.1$ and $\beta = 0.3$, Did the authors test other constants too? Are we sure there isn't a constant value of $\beta$ giving good performance?

**Limitations:**

The limitations are discussed in Appendix E. Maybe it would be better if moved to the main body of the paper.

**Strengths And Weaknesses:**

## Strenghts
1. The observation that output length should be modeled as a distribution rather than a point estimate is well-motivated for the LLM context, and the empirical validation of heavy-tailed behavior is convincing.
2. The paper provides comprehensive experiments across multiple scenarios (online/offline), datasets (LMSYS-Chat-1M, ShareGPT, Alpaca), and models (8B, 70B, ...), showing the practicality of their methods.
3. Thorough empirical evaluation: Extensive ablation studies on distribution families, hyperparameters, and design choices.

## Weaknesses
Unfortunately, I believe that the paper has some major weaknesses.
1. Missing fundamental literature: This paper fundamentally falls into the scope of stochastic scheduling and learning-augmented algorithms, yet I was very surprised to see that it completely fails to cite this extensive literature or compare against existing algorithms within it:
- 1.a) Literature o Stochastic scheduling: The paper cites only Weber (1983) on SEPT but the line of work on stochastic scheduling is very extensive and decades long, exploring scheduling with uncertain/random job durations, policies for heavy-tailed service times, robust scheduling under uncertainty, and queueing theory for systems with random service times (M/G/1 queues, etc.).
- 1.b) Literature on Learning-augmented algorithms: The paper proposes an uncertainty aware method to use predictions of the job lengths, and this falls exactly in the scope of algorithms with predictions. In particular, scheduling problems have been extensively studied within this framework. The learning-augmented paradigm also gives natural metrics to evaluate the performace of algorithms using uncertain predictions (consistency, robustness, smoothness).

This lack of comparison with existing literature makes it difficult to assess correctly the novelty or strength of the paper, and how its method's performance compares to SoTA algorithms of stochastic or learning-augmented scheduling. Therefore, it seems to me that a major revision is needed, and given the limited time for rebuttals, it is difficult to correctly do such extensive literature review, and correctly identify which of existing algorithms would perform best in this setting and do new experiments comparing against them. **This is mainly why I recommend rejecting the paper**.

2. The justification for using TIE as priority metric in the scheduling algorithm is limited. The paper explains why it makes sense to consider this metric (and I agree it does), but it is not clear to me why this specific formulation and not something else, for example $E[\tilde{X}] + \beta \text{Var}[\tilde{X}]$, $E[\tilde{X}] + \beta .\text{Std}[\tilde{X}]$, or $E[\tilde{X}] + \beta .\text{UCB}[\tilde{X}]$, where UCB is some upper confidence bound (inspired by bandits/RL algorithms), or combining with some other robustness metric than CVaR

3. The experimental section does not compare to any preemptive scheduling algorithms as baselines (although some were cited in the paper)

---

> ### Author Rebuttal · Authors · 2026-03-31
>
> We thank the reviewer for the careful comments.
>
> **W1. Missing literature on stochastic scheduling and learning-augmented algorithms.**
>
> We fully acknowledge that TIE falls within the scope of stochastic scheduling and learning-augmented algorithms, and that incorporating the relevant discussion helps clarify our positioning. However, the **gap** between theoretical algorithms and real-world LLM serving systems makes direct method comparison challenging, as detailed below:
>
> **1. Gap between classical theory and practice (which TIE aims to bridge).** As Mitzenmacher & Shahout (2025) [1] point out, applying theoretical results to real-world LLM systems still faces many open problems, such as resource contention, KV cache reuse and eviction, and prediction overhead (when a predictor is involved). Specifically, classical stochastic scheduling assumes that job size distributions are **fully known**, whereas in LLM inference, the output length distribution is not known a priori. In TIE, we show that output lengths can be effectively fitted with the log-t distribution, and train a predictor to estimate the distribution parameters, thereby making the distribution required by stochastic scheduling available in practice.
>
> **2. Gap between the M/G/1 model and batched serving.** Most algorithms in stochastic scheduling and queueing theory are built on the canonical **M/G/1 model**, which assumes that only one job is served at a time. However, continuous batching is already standard in modern LLM serving systems, where hundreds of requests undergo decoding simultaneously.
>
> **3. Current practice in the field.** The above gaps make it difficult to directly apply classical stochastic scheduling algorithms to LLM inference scheduling or to compare against them. Consequently, existing works (e.g., SSJF, LTR, S³, and ELIS) mostly discuss and compare against other implementations rather than theoretical results from classical scheduling.
>
> **4. Connection to learning-augmented algorithms.** This framework requires balancing performance when predictions are accurate *(consistency)* versus erroneous *(robustness)*. In TIE, the CVaR and adaptive β (in Eq. 12) **explicitly govern this trade-off**: when pressure is low, β stays small, favoring the predicted expectation *(consistency)*; as pressure increases, β grows, shifting weight toward tail risk *(robustness)*. The ablation study validates that this design outperforms both SEPT (consistency only, no robustness) and fixed-β variants (static trade-off).
>
> **W1-Summary.** We agree that discussing the relevant literature helps clarify our positioning. However, we view W1 as a positioning issue that can be resolved by incorporating the relevant discussion, rather than a technical limitation that diminish the novelty or contribution of our work. We will add these discussions to better position our work and would be grateful if you could reconsider your rating accordingly.
>
> **Q1+W2. Why CVaR?**
>
> In fact, we have systematically explored a wide range of score formulations, including combinations of E[X] with various metrics, multiple instantiations of the UCB framework, and robust optimization approaches. Due to the word limit, we provide the full results at \[[link1](https://anonymous.4open.science/r/TIE/Re/SF.pdf)\].
> Empirically, "E[X]+β·CVaR" achieves the best end-to-end performance among them. Theoretically, CVaR captures the conditional expectation over the **entire tail** of the distribution, providing a more comprehensive characterization of the risk of long outputs, which makes it particularly well-suited for the risk-sensitive scheduling principle.
>
> **Q2. Analysis for Adaptive β**
>
> We evaluate TIE with different fixed-β across RPS (Results in \[[link2](https://anonymous.4open.science/r/TIE/Re/Beta.pdf)\]). The best fixed β **shifts** with load (0.1 at 10 RPS, 0.3 at 30–60 RPS, 0.5 at 100–200 RPS). Even at 200 RPS, an excessively large β (0.7) over-emphasizes tail behavior and disrupts the overall order. Hence we set the range to [0.1, 0.5]. More importantly, real-world loads are **fluctuating** rather than static, making adaptive β essential for robust deployment.
>
> **W3. Comparison with Preemptive Scheduling**
>
> We compare with preemptive TRAIL (ICLR 2025) [2], showing **TIE outperforms TRAIL at moderate and high load**. (Results in \[[link3](https://anonymous.4open.science/r/TIE/Re/TR.pdf)\]).
>
> At low RPS (10–40), both methods are comparable, with TRAIL holding a slight edge. At moderate and high RPS (≥60), TRAIL degrades sharply, as frequent preemptions cause KV cache eviction and substantial re-prefill overhead. TIE reduces mis-ranking probability at the source through distributional modeling and risk-aware scoring, rather than costly preemption.
>
>  [1] Mitzenmacher, M., Shahout, R. Queueing, predictions, and large language models: Challenges and open problems. Stochastic Systems, 2025.
>
> [2] Shahout et al. Don't Stop Me Now: Embedding Based Scheduling for LLMs. ICLR 2025.

---

> > ### Author Rebuttal · Reviewer_9d7q · 2026-04-01
> >
> > I thank the authors for their response and for the additional results.
> >
> > As mentioned in my review, my main reason for recommending rejection was the lack of sufficient comparison with existing literature, which made it difficult to fully assess the novelty of the work. Following the authors’ rebuttal, I now have a clearer understanding of how the paper is positioned within the literature and why direct comparisons with prior work in stochastic scheduling are challenging. I nevertheless encourage the authors to incorporate this discussion into the paper.
> >
> > Moreover, the additional experimental results also clarify the choice of CVaR as a robustness metric and better illustrate the role of a dynamic $\beta$. These aspects previously appeared to me somewhat arbitrary, but are now much better motivated.
> >
> > I will raise my score to "weak accept" :)

---

> > > ### Author Response · Authors · 2026-04-07
> > >
> > > We greatly appreciate the reviewer for the thoughtful evaluation and for raising the score.
> > >
> > > We will incorporate a more detailed discussion on the positioning of our work within the stochastic scheduling and learning-augmented algorithms literature in the revised paper.

---

### Official Review · Reviewer_ctJh · 2026-03-12

**Soundness:** 3
**Presentation:** 3
**Significance:** 3
**Originality:** 3
**Overall Recommendation:** 5
**Confidence:** 3

**Summary:**

This paper argues that point estimates of output length are mismatched with the stochastic decoding process of LLMs, and proposes distributional modeling instead. The authors show that output lengths follow heavy-tailed distributions well-fitted by the log-t family (93.1% KS pass rate), prove power-law tail decay under mild assumptions (Theorem 3.2), and design TIE (Tail Inflated Expectation) — a scheduling score combining the censored expected length with a CVaR tail-risk term weighted by adaptive system pressure. A DeBERTa-v3-base predictor estimates log-t parameters (μ, σ) per prompt, integrated into vLLM via asynchronous prediction. Across three datasets, up to eight models (including MoE), and both online/offline scenarios, TIE reduces per-token latency by a factor of 2.31 over LTR and improves offline throughput by a factor of 1.42.

**Compliance With Llm Reviewing Policy:**

Affirmed.

**Final Justification:**

The authors delivered a thorough rebuttal that addressed all five key questions with new experimental evidence rather than qualitative arguments.

The paper's core contribution was already clean and well-evaluated in the original submission. The rebuttal has now closed the gaps I identified: the method is robust to prediction error, generalises across prompt families and sampling configurations, outperforms a preemptive baseline where it counts, and has practical training costs. The systems engineering (async prediction, dynamic batching, starvation prevention) remains a strength. I raise my score from 4 to 5.

**Key Questions For Authors:**

**Q1.** How sensitive is TIE to σ prediction quality? At what noise level does TIE degrade to SEPT or worse? A demonstration of graceful degradation would increase my confidence in the robustness of the approach and could move my recommendation toward a stronger accept; evidence of brittleness would be a significant concern.

**Q2.** Can you provide indirect support for Assumption 3.1 beyond tail-fit statistics — e.g., diagnostics showing the tail mechanism is consistent across prompt families (structured vs. open-ended) or decoding settings? This would tighten the theory-to-practice link. Even partial evidence would resolve W5; absence would not lower my score but would leave a gap.

**Q3.** Have you considered a simplified preemptive baseline with measured overhead? Your qualitative arguments are reasonable, but numbers would make the positioning concrete. A favorable comparison would strengthen the significance claim; even showing comparable performance with lower overhead would be informative.

**Q4.** Does TIE generalize across sampling configurations (temperature, top-p)? What happens when test-time settings differ from training? Evidence of robustness here would address W6; if retraining is needed per configuration, it compounds the adoption cost concern in W3.

**Q5.** What happens if TIE gets 900K unique prompts with fewer repetitions, or baselines get only 45K prompts? This would disentangle distributional modeling from prompt diversity effects. A clear answer either way would help me assess the practical efficiency of the approach.

**Limitations:**

The authors honestly discuss the multi-generation data requirement (Appendix E). However, the practical cost relative to baselines that train on free production logs deserves more prominent treatment. Undiscussed: (1) sampling parameter sensitivity, (2) fairness under sustained load, (3) domain-specific applicability, (4) the gap between Assumption 3.1 and empirical verification. The impact statement is boilerplate.

**Strengths And Weaknesses:**

### Strengths

**S1 — Clean core insight (Soundness, Originality).** The mismatch between point-estimate schedulers and stochastic decoding is simple but compelling. Figure 1 makes it concrete: the same prompt yields output lengths from ~100 to ~2000 tokens across 256 generations. This distributional framing is a genuine contribution that prior SJF methods overlook.

**S2 — Principled theory (Soundness).** Theorem 3.2 cleanly explains heavy-tailed output lengths: if per-trajectory termination rates have density f(p) ~ c·p^{α−1} near zero, then P(L > n) ~ c·Γ(α)/n^α. The proof is standard but correct, and the assumption is physically intuitive (structured outputs sustain low EOS probability over long spans).

**S3 — Thorough evaluation (Soundness, Significance).** 8 models (including 3 MoE), 3 datasets, online and offline scenarios, cross-model/cross-dataset generalization. The heatmaps (Figure 5) visually confirm tighter output-length concentration. The ablation (Table 4) validates both the distribution family and score computation choices.

**S4 — Real systems contribution (Significance).** This is not just "fit a distribution and stop." Adaptive risk sensitivity, starvation prevention, asynchronous prediction decoupled from the main thread, dynamic batching for the predictor, and the elegant default of max_tokens as initial score for unpredicted requests — these reflect genuine deployment thinking and make the method production-ready.

**S5 — Strong practical gains (Significance).** 2.31× latency reduction over LTR at 100 RPS, 1.42× throughput for SDG. Resilience to workload spikes (3.68× latency increase from 30→100 RPS vs 6.17× for LTR) matters for production.

### Weaknesses

**W1 — σ predictor robustness is underexplored (Soundness).** R² = 0.76 for σ is mediocre, yet σ controls the tail shape that TIE's CVaR term depends on. Does TIE degrade gracefully when σ is poorly predicted, or can errors actively harm scheduling? A controlled noise experiment would clarify.

**W2 — Limited justification for adaptive β (Soundness).** A pressure-based heuristic is reasonable for a systems paper, but the specific functional form and bounds of Eq. 12 lack justification. The ablation only tests fixed β ∈ {0.1, 0.3} vs adaptive, with modest gains (0.71 → 0.67). A broader sensitivity analysis would strengthen confidence.

**W3 — Training data cost as adoption barrier (Soundness, Significance).** TIE needs 45K prompts × 20 generations to fit distributions, whereas baselines train on production logs directly. The claim of "equal training data scale" (900K samples each) obscures that TIE sees 20× fewer unique prompts at ~20× higher cost per sample. How this upfront cost amortizes deserves transparent discussion.

**W4 — Positioning gap on preemptive methods (Significance).** The paper defensibly scopes comparisons to non-preemptive baselines and argues preemption is expensive, but without at least one measured preemptive comparison the reader cannot calibrate the tradeoff. This is a positioning limitation, not a flaw in the method.

**W5 — Assumption 3.1 only indirectly supported (Soundness).** The assumption concerns a latent variable (per-trajectory termination rates near zero), not directly observable from output lengths. Current support via tail statistics and KS fits is reasonable but leaves a gap. Diagnostics showing consistency across prompt families or decoding settings would help.

**W6 — No sampling parameter sensitivity (Soundness).** Output length distributions shift with temperature/top-p/top-k. Whether TIE requires retraining per sampling configuration is unaddressed.

**W7 — Insufficient starvation analysis (Soundness).** The decay mechanism (γ = 0.9, τ = 30s) lacks fairness metrics. Under sustained bimodal load, does it prevent pathological tail latency for long requests?

---

> ### Author Rebuttal · Authors · 2026-03-31
>
> Thank you for your recognition and valuable suggestions!
>
> **Settings:** unless otherwise specified, all experiments are conducted on Llama-3-8B-Instruct with the LMSYS-Chat-1M dataset, reporting average per-token latency (Pt-Lat) (s/token) at 100 requests per second.
>
> **Due to the word limit, we put the detailed experimental results in [\[link\]](https://anonymous.4open.science/r/TIE/Re/ctJh.pdf).**
>
> **Q1+W1: Robustness to σ Prediction Quality**
>
> We conduct a controlled noise experiment, showing that the performance degradation is **gradual**.
>
> **Table 1.** Avg Pt-Lat under varying σ noise levels.
>
> |RPS|TIE ($σ_{noise}$=0)|TIE (0.1)|TIE (0.3)|TIE (0.5)|TIE (0.7)|SSJF|LTR|
> |-|-|-|-|-|-|-|-|
> |60|**0.41**|0.43|0.49|0.55|0.60|0.91|0.76|
> |100|**0.67**|0.77|1.05|1.34|1.66|1.95|1.55|
>
> At moderate load (60 RPS), adaptive β buffers the tail's influence, keeping performance stable. At high load (100 RPS), only under extreme noise (0.7×, i.e., scheduling with 0.3–1.7× of the predicted σ) does TIE (1.66) fall within the range of baselines. Note that '0.7×' is relative to predicted values, so the tolerance relative to ground-truth is even larger.
>
> The scale parameter σ captures output length **variability**, inherently harder to predict than the location parameter μ. Nevertheless, the impact on end-to-end performance is limited.
>
> **Q2+W5: Cross-Prompt-Family Diagnostics**
>
> We show that heavy-tailed output-length distributions are **universal** across prompt families. We classify the 1K×100 prompt data and find that every category exhibits positive skewness, high P90/P50 ratios, and >89% log-t KS pass rates (see **Table 2** in the above link). Differences across categories reflect only tail heaviness, not the structural presence of tails. Open-ended prompts exhibit heavier tails, consistent with their fewer generation constraints.
>
> **Q3+W4: Comparison with Preemptive Scheduling**
>
> We compare with preemptive TRAIL (ICLR 2025) [1], showing **TIE outperforms TRAIL at moderate and high load**. (Results in **Table 3** in the above link).
>
> At low RPS (10–40), both methods are comparable, with TRAIL holding a slight edge. At moderate and high RPS (≥60), TRAIL degrades sharply, as frequent preemptions cause KV cache eviction and substantial re-prefill overhead. TIE reduces mis-ranking probability at the source through distributional modeling and risk-aware scoring, rather than costly preemption.
>
> **Q4+W6: Sensitivity to Temperature**
>
> TIE trained under a temperature of 0.7 **generalizes well** across other settings. (Results in **Table 4** in the above link).
>
> At low temperature (≤0.4), reduced uncertainty dilutes distributional modeling's advantage, but TIE still leads. At moderate temperature (0.7–1.3), TIE performs best. At extreme temperature (≥1.6), all methods degrade due to severely increased uncertainty and frequent HOL blocking. However, such settings are rarely used in practice as they produce low-quality outputs [2].
>
> **Q5+W3: Training Data Cost**
>
> We evaluate TIE with predictors trained on varying data scales, confirming that TIE's data requirement is **practical**. (Results in **Table 5** in the above link).
>
> With only 15K×10 samples, TIE surpasses SSJF trained on 900K samples and approaches LTR. Baselines trained on 45K×20 data degrade due to reduced semantic diversity.
>
> As shown in Table 3 of manuscript, generating 150K samples takes ~3 hours on a single A6000, or <0.5 hours with 8-GPU data parallelism. Given TIE's strong generalization, this cost is modest to the lifetime of an LLM system.
>
> **W2: Analysis for Adaptive β**
>
> We evaluate TIE with different fixed-β across RPS (Results in **Table 6** in the link). The best fixed β **shifts** with load (0.1 at 10 RPS, 0.3 at 30–60 RPS, 0.5 at 100–200 RPS). Even at 200 RPS, an excessively large β (0.7) over-emphasizes tail behavior and disrupts the overall order. Hence we set the range to [0.1, 0.5]. More importantly, real-world loads are **fluctuating** rather than static, making adaptive β essential for robust deployment.
>
> Alternative monotonic forms for Eq. 12 (e.g., √-scaling, log-scaling) yield comparable results as the underlying principle is shared. We adopt the linear form for simplicity.
>
> **W7: Starvation Prevention**
>
> Our decay mechanism can effectively prevent starvation.
> **Theoretically,** the exponential decay ensures that any request's score decreases monotonically toward zero as waiting time grows, granting long requests sufficient priority to compete with new short ones even under sustained bimodal load.
> **Empirically**, the heatmaps in Figure 9 show that long requests are not systematically deferred, indicating no pathological tail latency in practice.
>
> We will incorporate the above results and analyses in the revision. Thank you for your thoughtful feedback!
>
> [1] Shahout et al. Don't Stop Me Now: Embedding Based Scheduling for LLMs. ICLR 2025.
>
> [2] Minh et al. Turning Up the Heat: Min-p Sampling for Creative and Coherent LLM Outputs. ICLR 2025.

---

> > ### Author Rebuttal · Reviewer_ctJh · 2026-04-01
> >
> > The authors delivered a thorough rebuttal that addressed all five key questions with new experimental evidence rather than qualitative arguments.
> >
> > The paper's core contribution was already clean and well-evaluated in the original submission. The rebuttal has now closed the gaps I identified: the method is robust to prediction error, generalises across prompt families and sampling configurations, outperforms a preemptive baseline where it counts, and has practical training costs. The systems engineering (async prediction, dynamic batching, starvation prevention) remains a strength. I raise my score from 4 to 5.

---

> > > ### Author Response · Authors · 2026-04-07
> > >
> > > We sincerely thank the reviewer for the thorough and constructive review and for acknowledging the value of our additional experimental results.
> > > We are grateful for raising the score and will further polish the paper to reflect the valuable suggestions.

---

### Decision · Program_Chairs · 2026-04-30

**Decision:**

Accept (regular)

**Comment:**

This paper proposes TIE, a scheduling framework for LLM inference that models output lengths as distributions (log-t family) rather than point estimates, combining censored expected length with a CVaR tail-risk term integrated into vLLM. The empirical results are strong and consistent

The reviewers converged to 5/4/5/4 after rebuttal. The one outlier (initial score 2) raised concerns about missing connections to stochastic scheduling and learning-augmented algorithms literatures, and questioned specific design choices. The rebuttal addressed all major concerns with new experiments, and all reviewers acknowledged resolution.

Residual weaknesses are minor: the stochastic scheduling and learning-augmented literature connection needs to be incorporated into the paper (not just the rebuttal), and the training data cost discussion should be moved to the main body.

Decision. The core contribution is clear, the evaluation is thorough, and the rebuttal closed the substantive gaps. I expect the authors to incorporate the promised literature discussion and rebuttal experiments into the final manuscript.